# LEARNING MASSIVELY MULTITASK WORLD MODELS FOR CONTINUOUS CONTROL 🦎

**Nicklas Hansen**$^{\star}$**, Hao Su**$^{\star\dagger}$**, Xiaolong Wang**$^{\star\dagger}$
$^{\star}$University of California San Diego, $^{\dagger}$Equal advising
{nihansen,haosu,xiw012}@ucsd.edu

## ABSTRACT

General-purpose control demands agents that act across many tasks and embodiments, yet research on reinforcement learning (RL) for continuous control remains dominated by single-task or offline regimes, reinforcing a view that online RL does not scale. Inspired by the foundation model recipe (large-scale pretraining followed by light RL) we ask whether a single agent can be trained on hundreds of tasks with online interaction. To accelerate research in this direction, we introduce a new benchmark with 200 diverse tasks spanning many domains and embodiments, each with language instructions, demonstrations, and optionally image observations. We then present *Newt*, a language-conditioned multitask world model that is first pretrained on demonstrations to acquire task-aware representations and action priors, and then jointly optimized with online interaction across all tasks. Experiments show that Newt yields better multitask performance and data-efficiency than a set of strong baselines, exhibits strong open-loop control, and enables rapid adaptation to unseen tasks. We release our environments, demonstrations, code for training and evaluation, as well as 200+ checkpoints.

**Webpage:** **https://www.nicklashansen.com/NewtWM**

## 1 INTRODUCTION

Learning a generalist control policy that can perform a wide variety of tasks is an ambitious goal shared by many researchers, and significant progress has already been made towards that goal (Jang et al., 2021; Reed et al., 2022; Brohan et al., 2023; Open X-Embodiment Collaboration et al., 2023; Black et al., 2024). However, the dominant approach among current efforts is to train a large policy with supervised learning on an enormous dataset of near-expert trajectories collected by *e.g.* human teleoperation.

This approach has two major drawbacks: *(i)* it greatly limits the amount of data available for training, and *(ii)* performance of the resulting policy is ultimately bounded by the quality of demonstrations. For these reasons, the community is increasingly turning to reinforcement learning (RL) for continuous improvement of large models. While this new paradigm of large-scale pretraining followed by light RL has led to impressive capabilities in game-playing (Baker et al., 2022; Vasco et al., 2025) and reasoning (OpenAI, 2024; Guo et al., 2025; Su et al., 2025), the continuous control community remains dominated by narrow training tasks (Tassa et al., 2018; Cobbe et al., 2019; Kostrikov et al., 2020; Hafner et al., 2023; Cheng et al., 2024; Joshi et al., 2025) or strictly offline regimes (Lee et al., 2022; Hansen et al., 2024; Park et al., 2025), reinforcing a view that RL from online interaction does not scale in this domain.

In this work, we challenge this assumption and ask the following question: can a *single* policy be trained with online RL on hundreds of control tasks at once? To answer this question, we in-

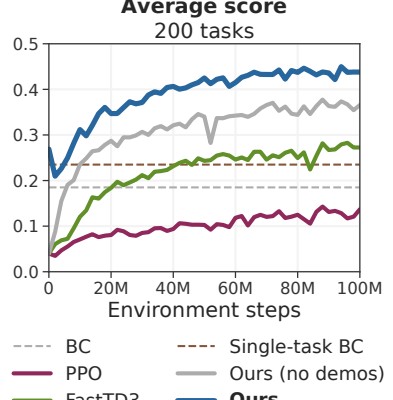

**Average score**
200 tasks

*Figure 1.* **Massively multitask RL.** Average score when training a *single* agent via online interaction on **200** tasks spanning **10** task domains.

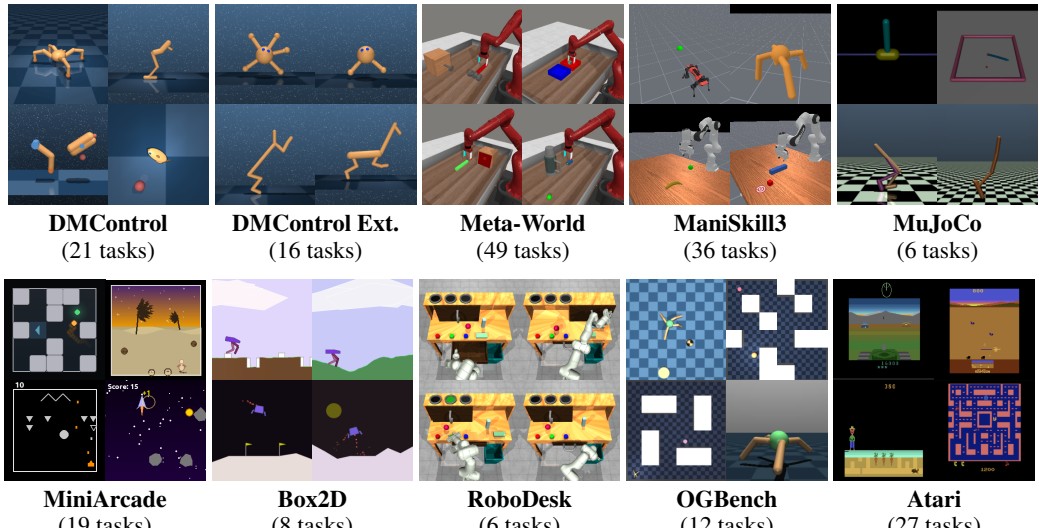

**DMControl**
(21 tasks)    **DMControl Ext.**
(16 tasks)    **Meta-World**
(49 tasks)    **ManiSkill3**
(36 tasks)    **MuJoCo**
(6 tasks)

**MiniArcade**
(19 tasks)    **Box2D**
(8 tasks)    **RoboDesk**
(6 tasks)    **OGBench**
(12 tasks)    **Atari**
(27 tasks)

*Figure 2.* **Tasks.** Our proposed benchmark, 🌍MMBench, consists of **200** distinct tasks across **10** task domains, including 41 new tasks. See Appendix C for a detailed overview of our task set.

troduce 🌍**MMBench**: the first benchmark for massively multitask RL. MMBench comprises of 200 diverse tasks spanning multiple domains and embodiments, each with language instructions, demonstrations, and optionally image observations, enabling research on both multitask pretraining, offline-to-online RL, and RL from scratch.

We then present 🦎**Newt**, a language-conditioned multitask world model based on TD-MPC2 (Hansen et al., 2024), which we first pretrain on demonstrations to acquire task-aware representations and action priors, and then jointly optimize with online interaction across all tasks. To extend TD-MPC2 to the massively multitask online setting, we propose a series of algorithmic improvements including a refined architecture, model-based pretraining on the available demonstrations, additional action supervision in RL policy updates, and a drastically accelerated training pipeline.

We validate our method on our proposed benchmark, and demonstrate that effective policy learning across hundreds of tasks is feasible with online RL and in fact can lead to strong multitask policies. Our experiments demonstrate that Newt *(1)* outperforms a set of strong baselines when training from state observations, *(2)* can be rapidly adapted to unseen tasks and embodiments by finetuning with online RL, *(3)* is capable of open-loop control over surprisingly long time horizons, and *(4)* benefits from access to high-resolution image observations. In support of open-source science, *we release 200+ model checkpoints, 4000+ task demonstrations, code for training and evaluation of Newt agents, as well as all 220 MMBench tasks (including 20 test tasks) considered in this work*.

In the following, we first introduce our benchmark MMBench, and then present our Newt world model. Readers can refer to Appendix A for a comprehensive discussion of related work on benchmarks, training infrastructure, and scaling of RL and control policies more broadly. We also provide recommendations for future work in Appendix B.

## 2 BENCHMARK FOR MASSIVELY MULTITASK REINFORCEMENT LEARNING

To study the feasibility of **M**assively **M**ultitask RL, we introduce 🌍**MMBench**: the first benchmark of its kind. MMBench contains a total of **200** unique continuous control tasks for training of massively multitask RL policies. The task suite consists of 159 existing tasks proposed in previous work, 22 new tasks and task variants for these existing domains, as well as 19 entirely new arcade-style tasks that we dub *MiniArcade*. The overarching goal of MMBench is to provide a common framework and infrastructure for research and prototyping of massively multitask RL. Figure 2 provides an overview of the **10** domains included in MMBench, Figure 3 shows tasks included in MiniArcade, and Appendix C provides a detailed description of each task domain. In the following, we discuss key features of our benchmark, as well as our efforts in making MMBench accessible to the broader research community.

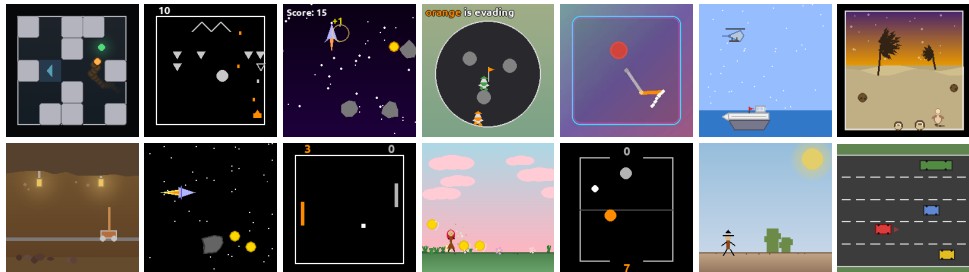

*Figure 3.* **MiniArcade.** We release a new task suite, dubbed *MiniArcade*, that consists of 22 tasks spanning **14** unique arcade-style environments (depicted). All tasks support both low-dimensional state representations and RGB observations, and have well-defined reward functions for RL.

> **Ant Ball** (OGBench)
>
> **Embodiment:** Ant (quadruped) with 8 controllable joints (4 legs).
> **Instruction:** Push the soccer ball to the goal location.

> **Rocket Collect** (MiniArcade)
>
> **Embodiment:** 2D space rocket with 2 controllable actions: main engine and side thruster.
> **Instruction:** Collect coins while avoiding asteroids.

*Figure 4.* **Sample language instructions.** All instructions in MMBench provide a description of embodiment and action space followed by a task description. Refer to Appendix D for more samples.

## 2.1 PROBLEM FORMULATION

Online reinforcement learning (RL) aims to learn a policy (agent) from interaction with an (in our case: multitask) environment. This interaction is commonly modeled as an infinite-horizon Partially Observable Markov Decision Process (Bellman, 1957; Kaelbling et al., 1998) formalized as a tuple $(\mathcal{S}, \mathcal{A}, \mathcal{T}, R, \gamma)$. Here, environment dynamics $\mathcal{T} : \mathcal{S} \times \mathcal{A} \mapsto \mathcal{S}$ are governed by generally unobservable states $\mathbf{s} \in \mathcal{S}$ approximated as $\mathbf{s} \doteq (s_{\text{state}}, s_{\text{img}}, s_{\text{lang}})$ where $s_{\text{state}}, s_{\text{img}}, s_{\text{lang}}$ are low-dimensional state inputs, image observations, and language instructions, respectively, $\mathbf{a} \in \mathcal{A}$ are actions, $\mathcal{R} : \mathcal{S} \times \mathcal{A} \mapsto \mathbb{R}$ is a reward function that produces task-specific rewards $r$, and $\gamma$ is a task-specific discount factor. Our overarching goal is to learn a *single* policy $\pi$ s.t. return $\mathbb{E}_{\pi} \left[ \sum_{t=0}^{\infty} \gamma^t r_t \right]$, $r_t = R(\mathbf{s}_t, \pi(\mathbf{s}_t))$ is maximized in expectation across all time steps and tasks, *i.e.*, a massively multitask policy trained to perform hundreds of tasks simultaneously via online RL.

## 2.2 ENVIRONMENTS

**Observations, actions, and rewards.** Our task suite comprises of diverse continuous control tasks spanning locomotion, tabletop manipulation, navigation, arcade games, classic control problems, and more. Tasks vary greatly in observation and action space dimensionality, task horizon, and reward specification. All tasks support three observation modes: *(1)* low-dimensional states, *(2)* $224 \times 224$ RGB images, or *(3)* both. To unify state observations and actions across tasks and domains, we provide a mask such that invalid dimensions can be taken into account during policy learning and inference. Unification of visual observations is done on a per-domain basis, with some domains readily supporting rendering at $224 \times 224$, while others require resizing (Box2D) or zero-padding (Atari). Refer to Appendix C for a table detailing the observations, actions, rewards, and episode lengths in each task domain.

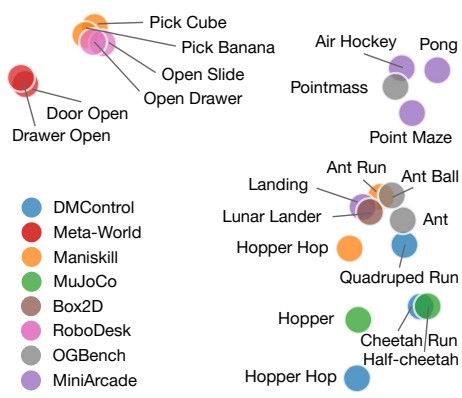

*Figure 5.* **Language embeddings.** First 2 principal components of `CLIP-ViT/B` embeddings shown for a subset of tasks.

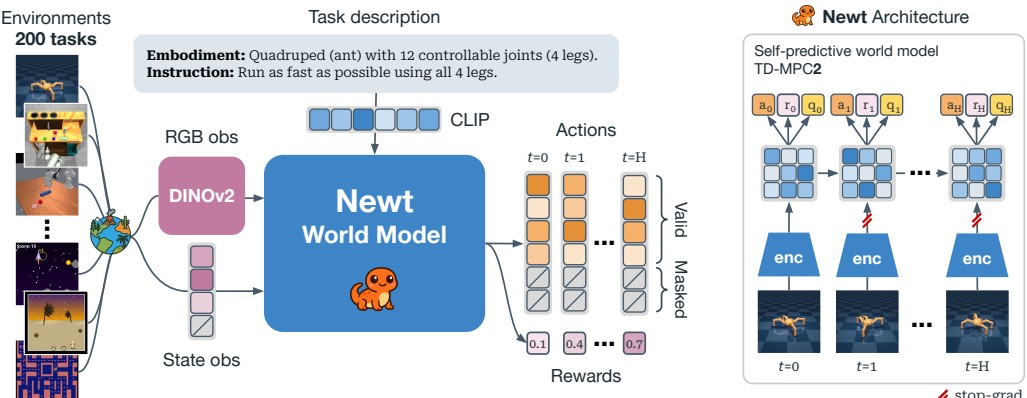

*Figure 6.* **Method.** Our agent (🦎) iteratively collects data via multitask environment (🌎) interaction, and optimizes its world model on the collected data. The world model takes a state vector, language instruction, and optionally RGB observations as input, and outputs actions via planning.

**Language instructions.** While some tasks can be differentiated solely based on observations, this is not universally true. For example, the manipulation tasks in *RoboDesk* share a common observation space (pose and velocity of robot and objects), and thus it is not clear from observations alone which object is to be manipulated. A simple solution would be to provide agents with a one-hot encoding of task indices, but this greatly limits the potential for transfer to new tasks since it provides no mechanism for representing unseen tasks. Instead, we choose to provide language instructions for every task in MMBench. Two sample language instructions are shown in Figure 4, with additional samples available in Appendix D. To verify that language instructions adequately differentiate tasks and embodiments, we encode all language instructions from MMBench with `CLIP-ViT/B`, and visualize their first two principal components in Figure 5.

## 2.3 ACCESSIBILITY

Training agents with online RL on a large number of tasks is challenging from a learning perspective and may be prohibitively expensive for researchers and practitioners with limited computational resources. To make MMBench accessible to the broader RL community, we have devoted significant time and effort to reduce computational costs.

**Demonstrations.** Exploration has historically been a key challenge in online RL, and a large body of literature is dedicated to exploration strategies (Brafman & Tennenholtz, 2003; Bellemare et al., 2016; Pathak et al., 2017; Sekar et al., 2020; Ecoffet et al., 2021; Ladosz et al., 2022). However, as problem scope and task complexity grows, learning an agent from scratch ("tabula rasa") via RL is becoming infeasible. Demonstrations serve as a strong behavior prior that dramatically reduces the exploration burden and variance of online RL, creating a practical path for researchers with limited compute to still contribute to the field. To this end, we provide 10-40 demonstrations for each task in MMBench, collected by single-task TD-MPC2 (Hansen et al., 2024) agents trained from low-dimensional state observations. While we do not directly leverage the trained single-task agents in this work (beyond collecting demonstrations), *we make all* 200 *model checkpoints available* for use by other researchers, and are excited to see what the community will use them for.

**Asynchronous environments.** Our benchmark consists of hundreds of environments in multiple robotics simulators, 2D game engines, and emulators, which complicates parallelization and overall software integration. To aid adoption, we provide a ready-to-use `docker` image and fast, easy-to-use environment wrappers that enable asynchronous environment stepping and rendering, batched frame-stacking and image encoding using large pretrained backbones, cached language embeddings, easy handling of diverse observation and action spaces, as well as auto-resetting whenever a task completes. These measures drastically reduce wall-time and allow users to get started *immediately*.

# 3 🦎 NEWT: A MULTITASK WORLD MODEL FOR CONTINUOUS CONTROL

To learn RL agents effectively in a massively multitask setting like ours, the algorithm of choice needs to satisfy the following criteria: it needs to *(i)* scale with increasing model and data size, *(ii)* be robust to various observation and action spaces, reward functions, and task horizons, *(iii)* be able to adequately differentiate tasks, and *(iv)* train in a reasonable time frame. We choose to base our agent, Newt, on model-based RL algorithm TD-MPC2 (Hansen et al., 2022; 2024) as it satisfies the first two criteria. Concretely, TD-MPC2 performs trajectory optimization (planning) in the latent space of a learned self-predictive (decoder-free) world model, and it has demonstrated robust learning across a variety of single-task (online RL) and multi-task (offline RL) environments. We extend TD-MPC2 to the massively multitask online RL setting, and describe our algorithmic improvements in the following. Figure 6 summarizes our approach.

## 3.1 LEARNING A MASSIVELY MULTITASK WORLD MODEL

TD-MPC2 learns its world model with a combination of joint-embedding prediction (self-predictive dynamics modeling), reward prediction, and TD-learning (Sutton, 1998); this is in contrast to generative world models that are typically trained to decode raw future observations (*e.g.* RGB images) using an auxiliary decoder network. A key benefit of self-predictive world models is, in addition to being computationally cheap due to not having a decoder, that they are trained to be *control-centric*: accurately predicting outcome (return) conditioned on a sequence of actions. Our proposed world model extends the TD-MPC2 architecture to support language instructions and optionally RGB observations in addition to low-dimensional state observations. Specifically, our world model consists of the following components:

$$
\begin{array}{llll}
\text{Language encoder} & \mathbf{g} = \text{CLIP}_{\text{text}}(\mathbf{s}_{\text{lang}}) & \triangleright \text{ Encodes natural language instruction} & \\
\text{Image encoder} & \mathbf{x} = \text{DINOv2}(\mathbf{s}_{\text{img}}) & \triangleright \text{ Encodes RGB image observation (\textit{optional})} & \\
\text{State encoder} & \mathbf{z} = h(\mathbf{s}_{\text{state}}, \mathbf{x}, \mathbf{g}) & \triangleright \text{ Computes latent state representation} & \\
\text{Latent dynamics} & \mathbf{z}' = d(\mathbf{z}, \mathbf{a}, \mathbf{g}) & \triangleright \text{ Predicts latent forward dynamics} & (1) \\
\text{Reward} & \hat{r} = R(\mathbf{z}, \mathbf{a}, \mathbf{g}) & \triangleright \text{ Predicts reward } r \text{ of a transition} & \\
\text{Terminal value} & \hat{q} = Q(\mathbf{z}, \mathbf{a}, \mathbf{g}) & \triangleright \text{ Predicts discounted sum of rewards (return)} & \\
\text{Policy prior} & \hat{\mathbf{a}} = p(\mathbf{z}, \mathbf{g}) & \triangleright \text{ Predicts optimal action } \mathbf{a}^* & \\
\end{array}
$$

where $\mathbf{s} = \{\mathbf{s}_{\text{lang}}, \mathbf{s}_{\text{img}}, \mathbf{s}_{\text{state}}\}$ are language, image, and state observations, respectively, $\mathbf{a}$ are actions. In practice, we use (frozen) pretrained backbones for language and image inputs which allows us to cache embeddings, and we choose to implement all other components as MLPs. Components that take multiple arguments as input fuse their inputs via concatenation before feeding them into the first dense layer, *i.e.*, we let $h(\mathbf{s}_{\text{state}}, \mathbf{x}, \mathbf{g}) \doteq h([\mathbf{s}_{\text{state}}, \mathbf{x}, \mathbf{g}])$ where $[\cdot]$ denotes concatenation.

Following TD-MPC2, we jointly optimize $h, d, R, Q$ by gradient descent on the objective

$$
\mathcal{L}(\theta) \doteq \mathbb{E}_{\tau \sim \mathcal{B}} \left[ \sum_{t=0}^{H} \lambda^t \left( \underbrace{\|\mathbf{z}'_t - \text{sg}(h(\mathbf{s}'_{\text{state}_t}, \mathbf{x}'_t, \mathbf{g}))\|_2^2}_{\text{Self-prediction}} + \underbrace{\ell_{\text{CE}}(\hat{r}_t, r_t)}_{\text{Rewards}} + \underbrace{\ell_{\text{CE}}(\hat{q}_t, q_t)}_{\text{Values}} \right) \right], \quad (2)
$$

where $\tau = (\mathbf{s}, \mathbf{a}, r, \mathbf{s}')_{0:H}$ is a subsequence sampled from a replay buffer $\mathcal{B}$, $\lambda \in (0, 1]$ is a constant coefficient which weighs temporally distant samples exponentially less, $\text{sg}$ is a $\text{stop} - \text{grad}$ operator that helps mitigate representation collapse (Grill et al., 2020), $\ell_{\text{CE}}$ is the cross-entropy loss, and predictions $(\mathbf{z}'_t, \hat{r}_t, \hat{q}_t)$ are as defined in Equation 1. Regressing rewards and values using a MSE loss is challenging in a multitask setting as reward distributions may be drastically different between tasks and task domains. Therefore, we opt for a discrete regression objective (the cross-entropy loss) and model values in a $\log$-transformed space such that we are able to model a wide range of values with a single prediction head (Bellemare et al., 2017; Kumar et al., 2023; Hafner et al., 2023; Hansen et al., 2024; Farebrother et al., 2024). We use $q_t = r_t + \gamma Q_{\text{tgt}}(\mathbf{z}'_t, p(\mathbf{z}'_t), \mathbf{g})$ as our one-step TD-target (Sutton, 1998), and let $Q_{\text{tgt}}$ be an exponential moving average (EMA) of the online $Q$ network (Lillicrap et al., 2016). In practice, we optimize a small ensemble of $Q$-networks and define the target $Q$-value as the minimum of a random subset of $Q_{\text{tgt}}$ estimates (Chen et al., 2021). We choose to use per-task discount factors ($\gamma$) since episode lengths vary drastically between tasks; we use the domain-default discount factor when one is available and otherwise estimate it using a heuristic described in Appendix H.

The policy prior $p$ is formulated as a stochastic maximum entropy policy (Ziebart et al., 2008) that learns to maximize $Q$-values as estimated by the $Q$-network defined above. However, we find that naive application of this policy objective to our problem setting leads to subpar performance in tasks where $Q$-values are difficult to estimate. Instead, we choose to leverage a small set of demonstrations (more on this in Section E) and add an additional behavior cloning loss to the policy prior of TD-MPC2. This serves two purposes: *(i)* directly leveraging expert demonstrations as action supervision, and *(ii)* explicitly distilling actions selected via planning into the less expressive policy prior. Concretely, we define the policy objective as

$$\mathcal{L}_p(\theta) \doteq \mathbb{E}_{\tau \sim \mathcal{B}} \left[ \sum_{t=0}^{H} \lambda^t \left[ \underbrace{\|p(\mathbf{z}_t, \mathbf{g}) - \mathbf{a}_t\|_2^2}_{\text{Model-based BC}} - \underbrace{Q(\mathbf{z}_t, p(\mathbf{z}_t, \mathbf{g}), \mathbf{g})}_{\text{Q-value}} - \underbrace{\mathcal{H}(p(\cdot|\mathbf{z}_t, \mathbf{g}))}_{\text{Entropy}} \right] \right], \quad (3)$$

where $\mathbf{z}_{t+1} = d(\mathbf{z_t}, \mathbf{a_t}, \mathbf{g})$, $\mathbf{z}_0 = h(\mathbf{s}_{\text{state}_0}, \mathbf{x}_0, \mathbf{g})$ is the latent rollout of $\tau$. During environment interaction, TD-MPC2 selects actions by planning with the learned world model, with the planning procedure warm-started by the policy prior $p$. Refer to Appendix G for details on our planning algorithm. In the following, we describe other ways in which we leverage demonstrations.

## 3.2 LEVERAGING DEMONSTRATIONS FOR WORLD MODEL LEARNING

To overcome the difficulty of exploration in massively multitask online RL, we choose to leverage a small number of demonstrations for each task. Although one could naively add demonstrations to the replay buffer at the start of training and indeed benefit from the model-based BC term introduced in Equation 3, we would like to utilize demonstrations to their full extent. Concretely, we propose to use demonstrations in four distinct ways:

*(1)* **Model-based pretraining.** Prior to any online interaction, we first pretrain all learnable components from Equation 1 on the provided demonstrations. Specifically, we assume that demonstrations consist of $(\mathbf{s}, \mathbf{a}, r)$ tuples and jointly optimize all components by minimizing $\mathcal{L}(\theta) + \mathcal{L}_p(\theta)$, but with the $Q$-value term in Equation 3 temporarily disabled such that we can fully leverage the strong action supervision from the demonstrations. This is in contrast to prior work (Zhan et al., 2020; Hansen et al., 2023a) that pretrains only encoder and/or policy.

*(2)* **Constrained planning.** When transitioning from pretraining to online RL, we empirically observe that (at the start of RL) planning with the world model yields a weaker behavior policy than directly using the pretrained policy due to a (comparably) inaccurate value function. To maximally retain performance when switching to planning, we initially bias the planner towards the pretrained policy and linearly anneal this bias to zero during the first 12% of training. See Appendix G for details.

*(3)* **Oversampling of demonstrations.** We maintain separate replay buffers for demonstrations and online interactions, and sample subsequences at equal proportions (50% from each) during agent updates (Feng et al., 2023; Ball et al., 2023). This means that demonstrations are (artificially) overrepresented in the training data, and ensures that the demonstration data remains available to the agent throughout training regardless of the capacity of the online interaction buffer.

*(4)* **Action supervision in RL policy updates.** As discussed in Section 3.1, adding a (model-based) BC loss term in the policy objective of Equation 3 provides direct action supervision and helps regularize the RL-based policy objective when $Q$-value estimation is inaccurate (Lin et al., 2025).

In summary, our Newt agent is a model-based RL method for language- and optionally image-conditioned massively multitask continuous control. It aims to fully leverage available demonstrations as well as online interaction data, resulting in a data-efficient yet computationally inexpensive method. In addition to the algorithmic improvements discussed in this section, we also **drastically accelerate training speed** by distributing both model updates, environment interactions, and replay buffers across multiple processes and GPUs, compiling training and inference code with `torch.compile` (Bou et al., 2023), and other implementation details described in Appendix H.

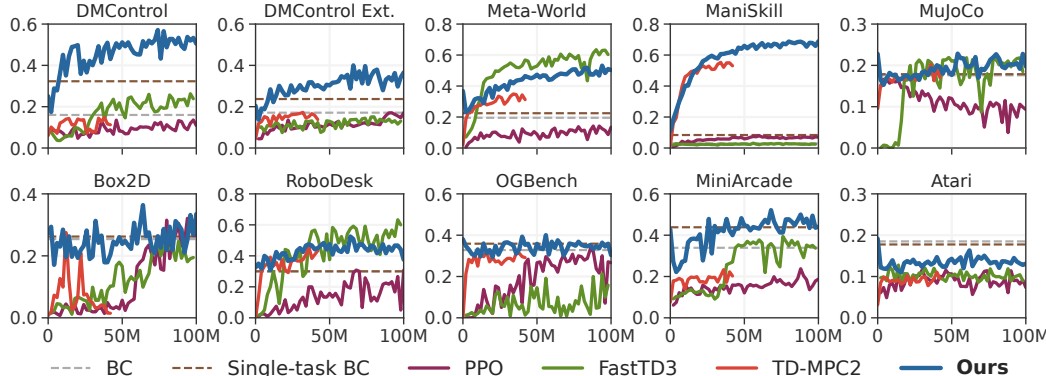

*Figure 7.* **Per-domain performance.** Average score of a *single* state-based agent on MMBench (10 task domains; 200 tasks). See Appendix J for more baselines, and Appendix K for per-task curves.

## 4 EXPERIMENTS

We evaluate 🦎**Newt** on our proposed benchmark 🌐**MMBench** which consists of **200** tasks across **10** task domains: DMControl (Tassa et al., 2018), DMControl Extended, Meta-World (Yu et al., 2019), ManiSkill3 (Tao et al., 2025), MuJoCo (Todorov et al., 2012), MiniArcade (a contribution of this work), Box2D (Brockman et al., 2016), RoboDesk (Kannan et al., 2021), OGBench (Park et al., 2025), and Atari (Bellemare et al., 2013; Farebrother & Castro, 2024). See Figure 2 for a visual overview of task domains, and refer to Appendix C for a detailed description of each task domain. Although our method can readily be applied to visual RL, experiments in this section use state observations unless we explicitly state otherwise. In support of open-source science, *we publicly release 200+ model checkpoints, 4000+ task demonstrations, code for training and evaluation of Newt agents, as well as all 220 MMBench tasks considered in this work.* We seek to answer:

(***Q1***) **Performance.** Can a *single* agent be trained on hundreds of unique tasks with online RL? How does our model-based approach (Newt) compare to a set of strong baselines?

(***Q2***) **Learning from demonstration.** Do demonstrations alleviate the difficulty of exploration in a massively multitask setting? How do we leverage demonstrations effectively for model-based RL?

(***Q3***) **Model capabilities.** What are the downstream capabilities of a massively multitask world model? Can we leverage our trained model for zero-shot or few-shot transfer to unseen tasks/embodiments? Can we perform open-loop control?

(***Q4***) **Analysis & ablations.** What makes a good multitask world model? What role does language and vision play? What are the current capabilities and limitations of Newt?

**Baselines.** Our baselines represent the state-of-the-art in data-efficient RL, learning from demonstrations, and RL with large amounts of parallel environments. Specifically, our baselines include:

• **Behavior cloning** (BC) (Pomerleau, 1988; Atkeson & Schaal, 1997) on demonstrations. We compare against a language-conditioned multitask BC policy, as well as • **200 single-task BC policies**.

• **Proximal Policy Optimization** (PPO) (Schulman et al., 2017) is an on-policy actor–critic method that maximizes a clipped surrogate objective, and it has gained popularity in part due to its fast wall-time training when combined with massive parallel simulation. We base our experiments on the `cleanRL` (Huang et al., 2022b) implementation, and extend it to support language-conditioning and per-task discount factors. We also tune hyperparameters; see Appendix J for results before/after.

• **FastTD3** (Seo et al., 2025), a modern implementation of TD3 (Fujimoto et al., 2018) designed for RL with parallel environments. We extend FastTD3 to support language-conditioning and per-task discount factors, and find that $n$-step returns of $8$ are critical in our challenging multitask setting.

• **TD-MPC2** (Hansen et al., 2022; 2024) trained with multitask online RL. This baseline can be considered a more naive implementation of Newt without language-conditioning, pretraining, demonstrations, nor the BC loss term introduced in Equation 3. We match the parameter count of Newt.

• **200 single-task TD-MPC2** (Hansen et al., 2022; 2024) agents trained on individual tasks for 5M environment steps (1B total steps). We collect demonstrations for BC and Newt using these agents.

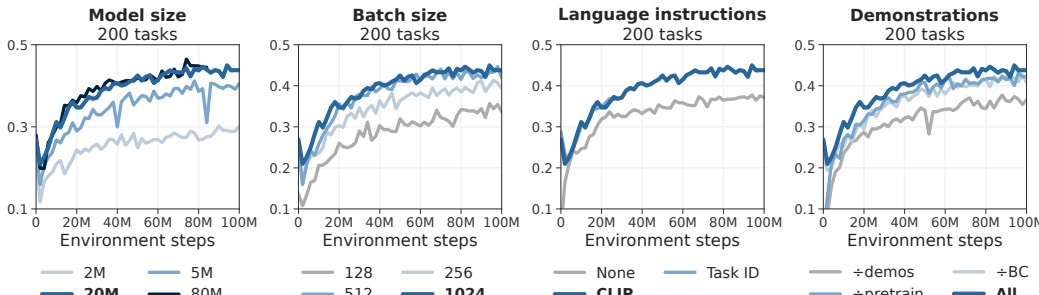

*Figure 8.* **Ablations** of all key design choices. Our default formulation of Newt is shown in **bold**.

• **Model-based pretraining** of our Newt world model on the same demonstrations as for BC. During pretraining, we optimize the model as described in Section 3.2.

See Appendix F for more information on baselines, including implementation and hyperparameters.

**Implementation details.** Language is encoded using `CLIP-ViT/B` (Radford et al., 2021) and images using `DINOv2/B` (Oquab et al., 2023), which results in embeddings of dimensions 512 and 768, respectively. State observations are 128-dim vectors, actions are 16-dim vectors, and our Newt agents have 20M learnable parameters unless stated otherwise. See Appendix H for more details.

## 4.1 RESULTS

**Benchmarking algorithms.** We evaluate the performance of our method, Newt, and baselines on our proposed MMBench task suite. Methods are trained for 100M environment steps (in total across all tasks) using low-dimensional state observations. Our main result is shown in Figure 1, and per-domain scores are shown in Figure 14. These results indicate that *Newt is more data-efficient and achieves a higher overall performance* than PPO, FastTD3, and TD-MPC2, which we attribute in part to its significantly better performance in *DMControl*, *DMControl Ext.*, *ManiSkill*, and *MiniArcade*. However, we also observe subpar performance across all RL methods in the *MuJoCo*, *Box2D*, and *Atari* domains, with the performance of Newt often being similar to that of the simpler BC baseline. We conjecture that this may be due to the relative uniqueness of tasks in these domains; for example, many Atari games have little in common beyond their action space. While we observe that the addition of demonstrations in many cases leads to better asymptotic performance as it alleviates the difficulty of exploration, we recognize that there is still room for improvement; developing methods that yield more consistent improvement across tasks is thus an exciting future research direction. Please refer to Appendix J and Appendix K for additional experiments and baseline comparisons.

**Analysis & ablations.** We ablate all key design choices, including model and batch size, use of language instructions, as well as all the different ways in which we can leverage demonstrations. Ablations are conducted with 20M parameter agents on the full 200-task set. Our main ablations are shown in Figure 8; additional experiments and per-domain results can be found in Appendix J. We make the following observations:

*Table 1.* **Language *sometimes* inhibits zero-shot generalization.** Success rate with seen and unseen instructions in 10 unseen manipulation tasks. 100 trials per task.

|  | Instruction | Success (%) |
|---|---|---|
| **(Default)** | `Push <unseen>` | 0.3 |
| | `Push <cube>` | **21.0** |
| **(Default)** | `Pick <unseen>` | **10.5** |
| | `Pick <cube>` | 0 |

*(1) **Scaling model and batch size is beneficial when the number of training tasks increases.*** In contrast to previous work that shows only marginal improvements from model scaling in a single-task RL setting (Hansen et al., 2024), we see a clear benefit in scaling model *and* batch size in a multitask setting, up to a point; see Appendix H for details on scaling. We conjecture that there exists a compute-optimal (*model*, *batch*) size for any given number of tasks in which learning is stable, and that further scaling of training tasks will require proportionally larger models and batches than presently.

*(2) **Language helps differentiate tasks.*** We find that conditioning the agent on language instructions provides clear performance benefits (0.371 → 0.438 normalized score), with the greatest improvement in domains where tasks cannot be differentiated by observations alone (*e.g.* RoboDesk); see

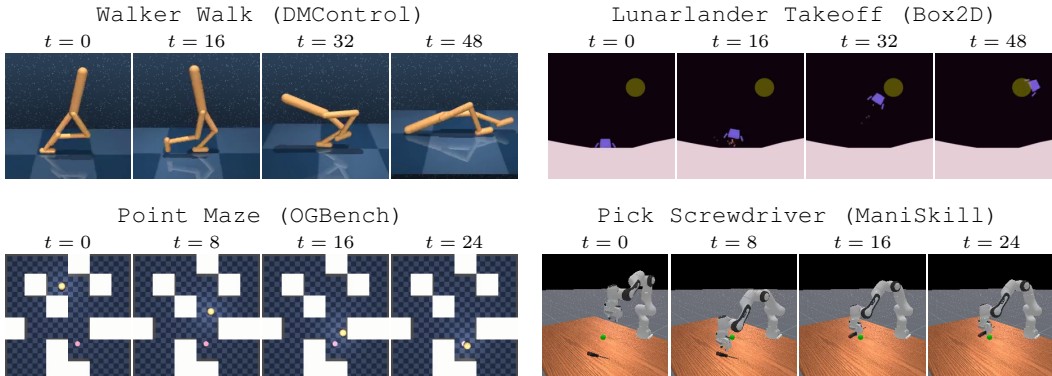

*Figure 10.* **Open-loop control.** Executing open-loop plans without any environment feedback. These results indicate that our world model learns meaningful representations of the environment.

Appendix J for a detailed performance comparison of Newt with and without access to language instructions. Additionally, as shown in Appendix L, this difference in downstream performance also translates to a lower training loss for the language-conditioned agent. In fact, we find language conditioning to match the performance of task indices (as used in TD-MPC2) on training tasks while *also* providing a mechanism for generalization to unseen tasks.

*(3)* ***Demonstrations improve performance in hard exploration tasks.*** We find that any individual way of using demonstrations (pretraining, oversampling, model-based BC loss) is helpful, but that using them all in conjunction yields the best data-efficiency and asymptotic performance.

**Task transfer.** To investigate whether our multitask agent transfers to unseen tasks/embodiments, we develop a held-out task set that spans multiple task domains and finetune our agent to each task individually using online RL and no demonstrations. Transfer tasks and aggregate finetuning results are shown in Figure 9. We observe that the pretrained Newt agent achieves a zero-shot score of **0.192** compared to 0.013 when trained from scratch, and reaches an average score of **0.868** at 100k environment steps vs. just 0.480 for the baseline. These results demonstrate non-trivial transferability, and we expect transfer results to get better as more and more training data becomes available. As shown in Table 1, we find that unseen language instructions can greatly inhibit zero-shot generalization of our agent, depending on the task. Replacing the noun describing the object to be manipulated (unseen instruction) with `cube` (inaccurate but seen instruction) improves zero-shot success rate by 20.7% across 6 pushing tasks, whereas we see the reverse trend for pick-and-place tasks. To best reflect the *true* capabilities of our agent, *we use unseen instructions* in all transfer experiments.

**Few-shot finetuning**
(20 unseen tasks/embodiments)

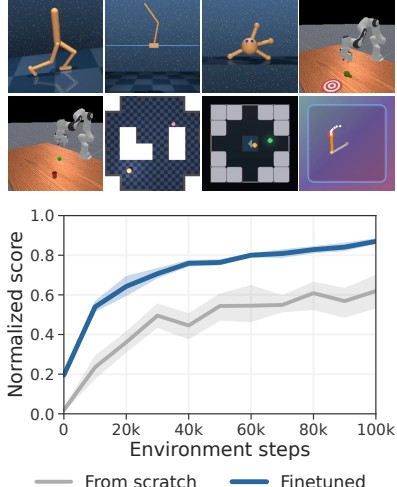

*Figure 9.* **Few-shot finetuning.** Average score when transferring our agent to new tasks/embodiments. 5 seeds.

**Open-loop control.** Model-based approaches are uniquely positioned to perform open-loop control (planning and executing actions without environment feedback). To better understand the current capabilities of Newt, we compare its open-loop planning performance relative to the default closed-loop planning. We evaluate on 8 diverse tasks and with planning horizons of **up to 48 time steps – 16× longer** than its training horizon of 3. Qualitative results are shown in Figure 10, and the full quantitative results are provided in Appendix I. We find that Newt can plan over long time horizons without being explicitly trained to do so, with performance closely matching closed-loop control in most tasks. Common failure modes include drifting dynamics (*Walker Walk*, DMControl), failing to decelerate after reaching the target (*Lunarlander Takeoff*, Box2D), or the inability to predict stochastic elements (*Assault*, Atari). Refer to Appendix I for more open-loop results.

**Visual observations.** The majority of our experiments are conducted with low-dimensional states due to (*i*) added computational costs of visual RL, and (*ii*) lack of available baselines for massively multitask visual RL. However, we hypothesize that some task domains stand to benefit more from vision than others. To test this hypothesis, we add $224 \times 224$ RGB inputs to our state-based agent as described in Equation 1 and finetune the entire model for 30M environment steps. We report average score across all domains as well as for a select few domains where the score change is noteworthy. We observe only a marginal overall improvement ($+0.004$) with visual observations, but find that performance *improves* significantly for manipulation (RoboDesk and Meta-World) and *decreases* in domains such as DMControl where learning from vision is known to be more challenging than from state (Hafner et al., 2019; Srinivas et al., 2020; Kostrikov et al., 2020).

**Training cost.** Table 3 shows the expected wall-time when training on 200 tasks from MMbench, for various hardware configurations and observation modes (state and state+RGB with rendering at $224 \times 224$). Machines are equipped with an AMD EPYC 9354 CPU and $\geq 128$ GB of RAM. The demonstration dataset requires 32 GB of disk space.

*Table 2.* **Visual RL.** Agent performance after finetuning with visual inputs for 30M steps.

| | Domain | Score | Gain |
|---|---|---|---|
| State | **All** | 0.438 | − |
| +RGB | **All** | **0.442** | +**0.004** |
| ↳ | RoboDesk | 0.500 | +**0.125** |
| ↳ | Meta-World | 0.572 | +**0.069** |
| ↳ | MiniArcade | 0.437 | −**0.004** |
| ↳ | DMControl | 0.471 | −**0.029** |

*Table 3.* **Training cost.** Expected wall-time when training on 200 tasks for 100M environment steps in total (20M params). Reported for different hardware configurations.

| Observation | Hardware | Days |
|---|---|---|
| State | 1× RTX 3090 | 11.2 |
| State | 2× RTX 3090 | 7.3 |
| State | 2× RTX 5090 | 4.6 |
| State+RGB | 2× RTX PRO 6000 | 4.7 |

## 5  CONCLUSION

We present MMBench, a benchmark for massively multitask RL, as well as Newt, a language-conditioned multitask world model trained with online RL. We demonstrate that online RL on hundreds of tasks simultaneously is feasible, and that it can lead to world models with surprisingly strong generalization and open-loop control capabilities. While we recognize that some challenges remain (rate of improvement may differ substantially between tasks and domains, generalization is currently inhibited by poor language understanding, as well as the significant hardware requirements for fast visual RL), we remain very optimistic about the future of massively multitask RL training.

**To encourage further research in this area**, we provide a comprehensive discussion of related work in Appendix A as well as recommendations for future work in Appendix B.

ACKNOWLEDGMENTS

The authors thank the following people for helpful discussions and feedback on paper drafts, in alphabetical order: Adrian Remonda, Arth Shukla, Bo Ai, Hansen Lillemark, Jiajun Xi, Lars Paulsen, Stone Tao, and Yutao Xie. A special thanks is also extended to the open-source RL community, without whom this project would not have been possible; particularly the original developers of MuJoCo, DMControl, Meta-World, ManiSkill3, RoboDesk, OGBench, Arcade Learning Environment, Gym and Gymnasium, TorchRL (Vincent Moens), as well as their numerous individual contributors.

This project was supported, in part, by NSF CAREER Award IIS-2240014 and NSF CCF-2112665 (TILOS), as well as gifts from Amazon, Meta, and Qualcomm.

## STATEMENTS

**A statement on reproducibility.** Reproducibility is important to us. We release 200+ model checkpoints, 4000+ task demonstrations, code for training and evaluation of Newt agents, as well as all 220 MMBench tasks (200 training tasks and 20 test tasks) considered in this work; all of our artifacts are made publicly available at **https://www.nicklashansen.com/NewtWM**. We also provide extensive implementation details and empirical results in the appendices. Most notably, Ap-

pendix C provides an overview of task domains, Appendix E provides details on demonstrations, Appendix F and Appendix H provide implementation details for baselines and Newt, respectively, and Appendix K provides learning curves for all 200 tasks.

**A statement on ethics.** We propose a benchmark, MMBench, that is comprised of both entirely new tasks, tasks that are derivative work based on existing task domains, and existing tasks that are used as is. All task domains and baselines for which we rely on third-party code are open-source and have permissible licenses. For example, DMControl is licensed under an Apache 2.0 License, OGBench is licensed under an MIT License, and Arcade Learning Environment (ALE; also known as Atari) is licensed under a GNU General Public License v2.0. Any derivative work of existing task domains is licensed under their respective licenses; code contributed as part of our work (*e.g.*, MiniArcade and Newt) is licensed under an MIT License.

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

APPENDICES

Webpage: **https://www.nicklashansen.com/NewtWM**

✨ **Related work:** Appendix A
✨ **Recommendations for future work:** Appendix B
🌍 **MMBench:** Appendices C−F
🦎 **Newt:** Appendices G−H
✨ **Additional results:** Appendices I−L

# A  RELATED WORK

Our work spans multiple research topics including *(i)* the development of new training environments, *(ii)* benchmarks for evaluation of multitask policies, *(iii)* algorithms and infrastructure for large-scale RL, and *(iv)* learning from demonstration. In the following, we aim to provide a comprehensive yet concise overview of related work along each of these axes.

**Benchmarks for multitask RL.** Historically, most RL benchmarks have been designed for single-task training and evaluation (Bellemare et al., 2013; Brockman et al., 2016; Tassa et al., 2018; Cobbe et al., 2019; Tao et al., 2025). While some benchmarks evaluate policy learning and generalization within a narrow task (Cobbe et al., 2019; Hansen & Wang, 2021; Kirk et al., 2023) or task family (Kolve et al., 2017; Yu et al., 2019; Savva et al., 2019; Kannan et al., 2021; Liu et al., 2023; Shukla et al., 2024; Park et al., 2025; Joshi et al., 2025), multi-domain learning and generalization in the context of RL remains mostly unexplored. Similarly, several works propose multi-embodiment datasets for imitation learning in robotics (Open X-Embodiment Collaboration et al., 2023; Li et al., 2024; Khazatsky et al., 2024) but we have yet to see a comparably large initiative for multitask and multi-*embodiment* RL, let alone multi-*domain* RL. Our proposed benchmark 🌎**MMBench** consists of 10 distinct task domains, each of which contains a variety of different tasks and embodiments that all have reward functions, demonstrations, language instructions, and optionally visual observations.

**Scaling RL.** Existing literature has predominantly explored scaling of control policies in the context of imitation learning, where access to large demonstration datasets for supervised policy learning (*e.g.* behavior cloning) is assumed (Jang et al., 2021; Reed et al., 2022; Schubert et al., 2023; Brohan et al., 2023; Open X-Embodiment Collaboration et al., 2023; Octo Model Team et al., 2024; Black et al., 2024). Perhaps most similar to ours, Reed et al. (2022) learns a multi-domain control policy via supervised learning on more than 63M demonstrations collected in various simulation environments. Although this is an impressive feat, relying on the availability of large amounts of expert demonstrations is highly impractical if not infeasible for many downstream applications, and the capabilities of the resulting agent is inherently limited by the behavior policy (*e.g.* a human or learned specialist policy) that generated the demonstrations. For this reason, the community is increasingly turning to RL for training and finetuning of agents with superhuman capabilities in game-playing (Berner et al., 2019; Schrittwieser et al., 2020; Lee et al., 2022; Baker et al., 2022; Vasco et al., 2025), reasoning and agentic AI (OpenAI, 2024; Guo et al., 2025; Su et al., 2025), and most recently robotics (Hafner et al., 2023; Hansen et al., 2024; Cheng et al., 2024; Miller et al., 2025; Nauman et al., 2025; DYNA Robotics, 2025). In particular, TD-MPC2 (Hansen et al., 2024) presents a model-based RL algorithm that can be trained on up to 80 tasks across 2 domains (DMControl and Meta-World) using offline RL on a large dataset of 545M transitions collected by specialist policies. However, RL still remains brittle to changes in learning dynamics (Fujimoto et al., 2018; Chen et al., 2021; Kostrikov et al., 2020; Hansen et al., 2023b; Farebrother et al., 2024), hyperparameters (Huang et al., 2022a; Hussing et al., 2024), and task specification (Clark & Amodei, 2016; Kirk et al., 2023) as evidenced by our ablations in Figure 8 and our zero-shot results in Table 1. Additionally, scaling of RL is often bottlenecked by training infrastructure and interaction throughput (Espeholt et al., 2019; Weng et al., 2022; Tao et al., 2025; Shukla, 2025; Seo et al., 2025). We provide a comprehensive benchmark for multi-domain RL accompanied by robust training infrastructure and a practical RL algorithm (🦎**Newt**) that consumes various data sources and modalities.

**Learning from demonstration.** Demonstrations and offline data have been used to bootstrap policies (Nakanishi et al., 2003; Ross et al., 2011; Ho & Ermon, 2016; Pinto et al., 2017; Duan et al., 2017; Peng et al., 2019; Kalashnikov et al., 2021; Baker et al., 2022; Ball et al., 2023), overcome the difficulty of exploration in sparse reward tasks (Vecerik et al., 2017; Hester et al., 2018; Rajeswaran et al., 2018; Zhan et al., 2020; Hansen et al., 2023a; Feng et al., 2023), and initialize multitask skills before online improvement with RL (Shi et al., 2022; Shukla et al., 2024; Zhang et al., 2024; Lu et al., 2025). By providing demonstrations for 200 tasks, MMBench provides a new, challenging testbed for massively multitask imitation learning, online RL, and offline-to-online RL in a common task suite that allows for fast, reliable, and reproducible measures of algorithmic improvement, and our agent Newt serves as a strong baseline for the community to build upon in this new paradigm. A research direction that we believe MMBench is especially suited for is finetuning of pretrained generalist models such as vision-language-action models (VLAs) using online RL as explored in Lu et al. (2025) for a narrower set of tasks than we consider in this work.

## B    RECOMMENDATIONS FOR FUTURE WORK

This work introduces MMBench, a benchmark for massively multitask RL, as well as Newt, a language-conditioned multitask world model trained with online RL. We demonstrate that online RL on hundreds of tasks simultaneously is indeed feasible, and that it can lead to world models with surprisingly strong generalization and open-loop control capabilities. We are excited by these encouraging results, and look forward to seeing in which ways the research community will build upon our work. In the following, we detail interesting open research questions and opportunities for future work in this area in hopes of inspiring new innovations. Specifically, we believe that the following 7 topics will have great potential for impact over the coming years:

- **Visual RL.** While MMBench and Newt fully support high-resolution ($224 \times 224$) RGB observations, the majority of our experiments in this work are limited to low-dimensional (128-d) state observations. Training visual RL policies has substantially higher hardware requirements, particularly so in terms of GPU memory if one is to store the replay buffer in GPU memory for fast sampling – and this is especially true in the massively multitask setting since it is necessary to retain some amount of data for each task. We believe that improving the practicality of visual RL remains an important research goal, and we see two future directions that could help alleviate the computational cost: *(1)* further pretraining of the visual encoder and world model on the provided demonstrations and/or large out-of-domain datasets, and *(2)* innovations in data pipelines that go beyond the typical first-in-first-out replay buffer.

- **Language understanding.** We find that language understanding is currently a bottleneck for task generalization despite leveraging pretrained language embeddings from CLIP (Radford et al., 2021). This may, in part, be due to the limited number of language instructions: one fixed instruction per task for a total of 200 tasks. We expect language understanding to improve as the number of training tasks increases, but we also believe that additional techniques such as data augmentation, pretraining on external datasets that contain language instructions, learning from language at the token-level rather than embeddings, or any other techniques that seek to improve the agent's ability to interpolate instructions will be immensely helpful.

- **Pretraining and improved base models.** Pretraining of the Newt agent is currently limited to supervised learning on a demonstration dataset. Our ablations show that improving the pretraining stage leads to consistently better performance both pre- and post-RL, so we expect further improvements and scaling of the model pretraining to be a valuable research direction. We expect training of agents for embodied decision-making and control to eventually resemble the training recipe of contemporary reasoning models (OpenAI, 2024; Guo et al., 2025) in which significant resources are invested into large-scale pretraining of an agent before any RL is performed.

- **Neural architectures.** Our Newt agent is based on the TD-MPC2 (Hansen et al., 2024) architecture which has proven to perform well across a variety of tasks and model sizes. However, the architecture remains rather simple: each component of the world model architecture is a deterministic MLP (except for the Gaussian policy prior). We believe that leveraging more recent architectural innovations such the Transformer (Vaswani et al., 2017) and Diffusion Policy (**?**) could potentially lead to further performance improvements, but their integration into model-based RL algorithms remains relatively unexplored.

- **Model capabilities.** Our initial explorations into the capabilities of our Newt agent indicate that it is capable of open-loop control across hundreds of tasks, and that it can be rapidly adapted to new tasks and embodiments using online RL. However, further research into the emerging capabilities (and current limitations) of massively multitask agents is warranted. For example, the structure of the emerging (learned) latent state space and by extension dynamics modeling is not well understood, but we believe that improved understanding of emerging behavior in agents will help inform the design of future agents and neural architectures for model-based RL.

- **Learning strategies.** We find convergence rate to differ substantially between tasks and task domains. This is perhaps not surprising as tasks vary greatly in complexity, degree of randomization, and reward specification. While our experiments show that the provided demonstrations greatly improve data-efficiency as well as asymptotic performance of our Newt agent, we believe that more sophisticated training strategies may also lead to better data-efficiency and asymptotic performance. Two directions that appear particularly interesting are *(1)* learning curricula that dynamically balance environment interaction (data collection) for each task based on task

progress, and *(2)* non-uniform sampling methods or training objectives that prioritize sampling from tasks or subtrajectories that are particularly useful.

- **Training environments and datasets.** We believe that further scaling of the environments and datasets used for training of world models will continue to drive performance and lead to better generalization abilities. Development of additional RL environments using *e.g.* procedural generation, agentic AI, or other generative methods is thus a promising research direction, in addition to training techniques that incorporate existing large-scale datasets from other domains.

Overall, we remain very optimistic about the future of massively multitask RL training, and hope that our emphasis on transparency and sharing of code, data, and checkpoints will inspire other researchers to do the same.

—— **Appendices continue on the next page** ——

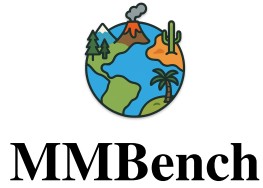

# MMBench

Appendices B−E

---

## C    TASK DOMAINS

We consider **200** tasks across **10** domains. Our task set is comprised of diverse continuous control tasks spanning robot manipulation, locomotion, navigation, arcade games, and classic control problems, each varying in task complexity, time horizon, observation and action space dimensionality, and reward formulation. Although we cannot provide detailed information about every task due to the sheer number of tasks considered, this section aims to give the reader a brief overview of the *types* of tasks used in our experiments. Table 4 provides an overview of the task domains considered.

*Table 4.* **Overview of task domains.** Our selection of tasks cover a wide range of task types, state and action dimensionalities, time horizons, and reward formulations. We summarize them below.

| Task domain | State dim | | Action dim | | Ep. length | | Reward | RGB |
| | min | max | min | max | min | max | | |
| --- | --- | --- | --- | --- | --- | --- | --- | --- |
| DMControl | 3 | 78 | 1 | 12 | 500 | 500 | dense/sparse | $224 \times 224$ |
| DMControl Ext. | 3 | 19 | 1 | 7 | 500 | 500 | dense/sparse | $224 \times 224$ |
| Meta-World | 39 | 39 | 4 | 4 | 100 | 100 | dense | $224 \times 224$ |
| ManiSkill3 | 10 | 42 | 1 | 12 | 25 | 500 | dense/sparse | $224 \times 224$ |
| MuJoCo | 4 | 105 | 1 | 8 | 50 | 1000 | dense/sparse | $224 \times 224$ |
| MiniArcade | 4 | 44 | 1 | 2 | 200 | 500 | dense/sparse | $224 \times 224$ |
| Box2D | 8 | 24 | 2 | 4 | 500 | 500 | dense | $224 \times 224$ |
| RoboDesk | 73 | 73 | 5 | 5 | 100 | 100 | dense | $224 \times 224$ |
| OGBench | 9 | 42 | 2 | 8 | 100 | 1000 | dense | $224 \times 224$ |
| Atari | 128 | 128 | 3 | 3 | 1000 | 1000 | sparse | $224 \times 224$ |

### C.1    DMCONTROL

DMControl (Tassa et al., 2018) is a benchmark for continuous control and features a variety of locomotion, manipulation, and control tasks with simple embodiments. It is available at `https://github.com/google-deepmind/dm_control`. All tasks are designed as infinite-horizon MDPs (no termination conditions), and all tasks have a fixed episode length of 500 when an action repeat of 2 is applied, as is standard practice for this benchmark (Yarats et al., 2021; Hafner et al., 2023; Hansen et al., 2024). Reward formulation varies between tasks; some tasks have dense (shaped) rewards whereas others are sparse and only provide a non-zero reward on success. DM-Control tasks do not have a notion of success, but episode returns are designed to be in [0, 1000] for every task, so we produce a normalized score for this benchmark by scaling the returns to the [0, 1] interval. Note that this does not guarantee that a score of 1.0 can be obtained in all DMControl tasks,

it is simply an upper bound. We consider **21** tasks from this domain. Examples of tasks included in our DMControl training set are *Finger Turn Hard*, *Fish Swim*, and *Quadruped Run*.

## C.2 DMCONTROL EXTENDED

DMControl Extended is, as the name implies, an extended task set based on the original DMControl environment. We include 11 custom tasks previously proposed by Hansen et al. (2024) (which are all available at https://www.tdmpc2.com), and design an additional 5 locomotion tasks across 3 new embodiments: *Jumper*, *Spinner*, and *Giraffe*. Tasks included in this task set follow the same recipe as the original DMControl benchmark: a fixed episode length of 500 steps, and episode returns in $[0, 1000]$ which we normalize by scaling down to $[0, 1]$. We consider **16** tasks from this domain. Examples of tasks included in our DMControl Extended training set are *Walker Run Backward*, *Cheetah Jump*, and *Spinner Spin*.

## C.3 META-WORLD

Meta-World (Yu et al., 2019) is a benchmark for robotic table-top manipulation tasks with a Sawyer robot and end-effector position control, and it is readily available at https://github.com/Farama-Foundation/Metaworld. It consists of 50 diverse manipulation tasks that all share a common observation- and action space to facilitate multitask learning. We use a fixed episode length of 100 steps (after applying an action repeat of 2), and no terminal conditions. Meta-World provides dense reward functions and success criteria for each task in the task suite; we report binary task success at end of episode as the normalized score for Meta-World tasks. We exclude one task, *Shelf Place*, from our training set due to an issue with the simulation, and consider the remaining **49** tasks from this domain. Examples of tasks included in our Meta-World training set are *Assembly*, *Bin Picking*, and *Lever Pull*.

## C.4 MANISKILL3

ManiSkill3 (Tao et al., 2025) is an open-source simulator and task suite for embodied AI. It features a wide range of tasks and embodiments including tabletop manipulation, quadruped locomotion, whole-body humanoid control, mobile manipulation, as well as reproductions of popular control tasks from MuJoCo and DMControl. Official documentation is available at https://maniskill.readthedocs.io. Observation and action spaces are defined on a per-task basis but is relatively consistent within a specific group of tasks. For example, most ManiSkill3 tabletop robotic manipulation tasks share a common observation space, but the action space depends on the specific robot embodiment and control mode. Robot embodiments in our ManiSkill3 task set include Franka Emika Panda, SO100, and xArm6 for manipulation, and Anymal for locomotion, in addition to simpler MuJoCo-like embodiments such as Ant, Hopper, and Cartpole. Control modes include end-effector position control, end-effector 6D pose control, and joint position control. We use the default episode lengths for each task but apply an action repeat of 2 and use no terminal conditions. ManiSkill3 provides dense reward functions and success critera for most tasks in the task suite, but not all. We use a dense reward function when one is available. We report binary task success at end of episode as the normalized score for ManiSkill3 tasks whenever one is available, and episode return normalized to the $[0, 1]$ interval otherwise. Note that this does not guarantee that a score of 1.0 can be obtained in all ManiSkill3 tasks, it is simply an upper bound. We consider a total of **36** tasks from this domain, of which 5 tasks are new and created specifically for our paper; we intend to submit a pull request to the official ManiSkill3 code repository with our proposed task additions. Examples of tasks included in our ManiSkill3 training set are *Stack Cube*, *Pick Screwdriver*, and *Anymal Reach*.

## C.5 MUJOCO

MuJoCo (Todorov et al., 2012) is an open-source physics simulator and classic benchmark in the reinforcement learning and continuous control research communities. Official documentation for the tasks is available at https://gymnasium.farama.org/environments/mujoco. Tasks in this suite are diverse in terms of embodiments and all have different observation and action spaces. We use the default episode lengths for each taks but apply an action repeat of 2 for some of the tasks.

We use the `v4` version of the tasks and disable all termination conditions to be in line with our other task domains. All tasks have pre-defined reward functions, some of which are sparse (only non-zero reward when success is reached), and we report episode return normalized to the $[0, 1]$ interval as normalized score for MuJoCo tasks. We consider a total of **6** tasks from this domain. Examples of tasks included in our MuJoCo training set are *Ant*, *Half-cheetah*, and *Reacher*.

## C.6 MINIARCADE

MiniArcade is a new task suite created specifically for this paper, implemented using `pygame` (https://github.com/pygame/pygame). It consists of 22 tasks spanning 14 unique arcade-style environments. Tasks vary greatly in task objective, episode length, observation and action space dimensionality, as well as reward formulation. All tasks have a fixed episode length and no termination conditions. We design each task to fully support low-dimensional state representations, RGB image observations, and a combination of the two. We define normalized return by pre-defined success criteria for tasks where it is appropriate to do so, and otherwise use episode return normalized to the $[0, 1]$ interval. As a result, there is no guarantee that a score of $1.0$ can be obtained in all MiniArcade tasks, it is simply an upper bound. We consider a total of **19** tasks from this domain for training, and have **fully open-sourced the entire task suite** for the community to build upon. Examples of tasks included in our MiniArcade training set are *Spaceship*, *Coconut Dodge*, and *Chase-Evade*.

## C.7 BOX2D

Box2D is a small collection of 2D environments created for the original release of OpenAI Gym (Brockman et al., 2016). Official documentation for the environments is available at https://gymnasium.farama.org/environments/box2d. The tasks vary in task objective, episode length, and observation and action space dimensionalities. We use a fixed episode length for every task by disabling termination conditions. The Box2D tasks that we consider were originally designed for low-dimensional state observations, so we have modernized the implementations and added support for visual observations, including (visual) domain randomization to promote more generalizable visual control policies. Tasks in this task suite have pre-defined dense reward functions that we use for policy learning, and we report normalized score as episode return normalized to the $[0, 1]$ interval. Note that this does not guarantee that a score of $1.0$ can be obtained in all Box2D tasks, it is simply an upper bound. We consider a total of **8** tasks and task variations based on the original OpenAI Gym environments; 5 for the *Bipedal Walker* environnment and 3 for the *Lunarlander* environment. Examples of tasks included in our Box2D training set are *Bipedal Walker (Obstacles)*, *Lunarlander Land*, and *Lunarlander Takeoff*.

## C.8 ROBODESK

RoboDesk (Kannan et al., 2021) is a small suite of robotic manipulation tasks designed for multitask RL research. It includes 9 object manipulation tasks in a single desk-themed environment, and all tasks share a common observation and action space. An implementation of RoboDesk is available at https://github.com/google-research/robodesk. The benchmark is explicitly designed for visual RL but also supports low-dimensional state observations. All tasks have a fixed episode length and no terminal conditions, and both dense reward functions for policy learning and success criteria for evaluation are provided; we use the success rate as normalized score for RoboDesk. We consider **6** tasks from this domain. Examples of tasks included in our RoboDesk training set are *Push Green*, *Open Drawer*, and *Place Flat Block in Bin*.

## C.9 OGBENCH

OGBench (Park et al., 2025) is a benchmark designed to facilitate research in offline goal-conditioned RL. It consists of multiple distinct environments and embodiments, and supports both low-dimensional state representations as well as RGB image observations. OGBench was developed with multi-goal but not multi-embodiment learning in mind, and the action space dimensionality thus depends on the particular task. Tasks vary greatly in episode length, but we use a fixed episode length for each task without any terminal conditions. Because OGBench was not designed for online

RL, we define dense reward functions for tasks where one was not previously defined, and modify the observation space to include all relevant task information (*e.g.* goal position). Unlike the original OGBench environments available at https://github.com/seohongpark/ogbench, we treat different goal positions within the same scene configuration as the same task. To increase environment difficulty and diversity, we create a series of novel scene configurations for the 2D point mass and Ant (quadruped) goal-conditioned navigation environments. We use success rate as normalized score for all but one task (*Ant* locomotion) which does not have a clear metric of success; we thus use episode return normalized to the $[0, 1]$ interval for this task. We consider **12** tasks based on the original OGBench environments; 5 tasks for the 2D point mass embodiment and 7 for the Ant embodiment. Examples of tasks included in our OGBench training set are *Ant Soccer*, *Point Mass (Maze)*, and *Ant (Bottleneck)*.

## C.10   ATARI

The Arcade Learning Environment (Bellemare et al., 2013), often referred to as simply Atari or ALE, is an interface for a collection of diverse and often challenging Atari 2600 games originally designed for human play. The ALE is a long-standing benchmark for RL algorithms, and has traditionally been used primarily for single-task visual RL benchmarking with a discrete action space. More recently, Farebrother & Castro (2024) has proposed a non-linear continuous-to-discrete action transformation that extends support to agents with continuous action spaces, which makes it a suitable task suite for our purposes. An implementation of ALE with support for continuous action spaces is available at https://github.com/Farama-Foundation/Arcade-Learning-Environment. However, several properties of ALE still make it challenging to apply directly to our problem setting: *(1)* the ALE is built for visual RL and unlike our other task domains it does not support low-dimensional state representations for each game; and *(2)* games vary by several orders of magnitude in terms of episode length, with a single episode in games like Assault sometimes exceeding $100,000$ steps with a human-level agent. To address the first challenge, we opt for the (admittedly not ideal) solution of using the raw game RAM state, which can be represented as a 128-dim vector, as our low-dimensional state representation. While we have verified that single-task agents are able to achieve non-trivial performance in all games considered, we recognize that using the raw RAM state as observation limits the potential for cross-task knowledge transfer. We therefore expect visual policies to have better potential for generalization in the Atari domain. To address the second challenge, we simplify our problem setting by limiting agents to a fixed episode length of $1,000$. Each Atari game varies greatly in their reward formulation, but games generally assign sparse rewards only at key events such as scoring a goal or losing a life. The games have no clear success criteria and we thus choose to use episode return normalized to the $[0, 1]$ interval as normalized score for this domain. The linear transformation between episode return and normalized score is defined for each task and takes the shorter episode length into account. However, this does not guarantee that a score of $1.0$ can be obtained in all Atari games, it is simply an upper bound. We consider **27** tasks (games) from this domain. Examples of tasks included in our Atari training set are *Ms. Pacman*, *Bowling*, and *Space Invaders*.

## D   SAMPLE LANGUAGE INSTRUCTIONS

We manually define language instructions for all 200 training tasks considered in this work. Our language instructions have two components: *(1)* a description of the embodiment, including how many controllable joints or action dimensions the given task has; and *(2)* a short task instruction in plain language. We make an effort to be consistent in our description of embodiments, action spaces, and tasks, while still providing enough language variation for generalization to be feasible. We provide a small subset of our language instructions in the following.

---

**Walker Run (DMControl)**

Embodiment: 2D walker with 6 controllable joints across 2 legs.
Instruction: Run forward as fast as possible.

---

**Walker Run Backward (DMControl Ext.)**

Embodiment: 2D walker with 6 controllable joints across 2 legs.
Instruction: Run backward quickly.

**Jumper Jump (DMControl Ext.)**

Embodiment: 2D jumper with 2 controllable joints across 2 legs.
Instruction: Jump as high as possible.

**Plate Slide (Meta-World)**

Embodiment: Sawyer robot with a 2-finger gripper and delta end-effector position control.
Instruction: Slide the hockey puck into the goal.

**Anymal Reach (ManiSkill3)**

Embodiment: Quadruped (Anymal robot) with 12 controllable joints across 4 legs.
Instruction: Reach target.

**Stack Cube (ManiSkill3)**

Embodiment: Franka robot with a 2-finger gripper and delta end-effector 6D-pose control.
Instruction: Stack the red cube on top of the green cube.

**Half-cheetah (MuJoCo)**

Embodiment: 2D MuJoCo half-cheetah with 6 controllable joints.
Instruction: Run forward as fast as possible.

**Bipedal Walker Obstacles (Box2D)**

Embodiment: 2D cartoonish bipedal walker with 4 controllable joints across 2 legs.
Instruction: Traverse the various obstacles quickly.

**Lunarlander Land (Box2D)**

Embodiment: 2D lunar lander with 2 controllable thrusters.
Instruction: Land safely on the landing pad.

**Open Slide (RoboDesk)**

Embodiment: Franka robot with a 2-finger gripper and delta joint position control.
Instruction: Open the sliding door.

**Ant Maze (OGBench)**

Embodiment: Ant (quadruped) with 8 controllable joints (4 legs).
Instruction: Navigate through a maze to reach the red goal.

**Bird Attack (MiniArcade)**

Embodiment: 2D shooter game with 2 controllable actions: move and fire.
Instruction: Eliminate all enemies in a space-invaders style game.

**Chase & Evade (MiniArcade)**

Embodiment: 2D chase and evade game with X and Y movement.
Instruction: Take turns being the chaser or evader.

**Ms. Pacman (Atari)**

Embodiment: Atari game with 3 controllable actions: radius, theta, and fire.
Instruction: Eat pellets and avoid ghosts in a game of Ms. Pacman.

# E  DEMONSTRATIONS

To aid in research on pretraining + finetuning of massively multitask agents we provide an expert demonstration dataset for MMBench. This dataset contains 10-40 demonstrations per task (depending on task difficulty) for a total of 4020 demonstrations, which are all collected by single-task TD-MPC2 (Hansen et al., 2024) agents trained from low-dimensional state observations. Each single-task agent is trained for 5M environment steps (Atari: 10M) to ensure that performance has converged, although we observe that training converges earlier than that on most tasks. Training a single-task agent for 5M environment steps takes approx. 6 hours on a single NVIDIA RTX 5090 GPU, totaling an estimated 1,300 GPU hours to train expert policies for every task in MMBench. To ensure that only successful demonstrations are collected, we reject episodes that do not result in success (when available) or have episode returns that are substantially ($> 25\%$) lower than the median for that task. Table 5 provides an overview of our demonstration dataset. For completeness, we also release all 200 checkpoints that generated the provided demonstrations; these checkpoints can be used to generate additional checkpoints, or be used for *e.g.* teacher-student policy distillation.

*Table 5.* **Demonstrations.** An overview of the demonstrations provided for each task domain.

| Task domain | Num. tasks | Demos per task | Total demos |
|---|---|---|---|
| DMControl | 21 | 20 | 420 |
| DMControl Ext. | 16 | 20 | 320 |
| Meta-World | 49 | 10 | 490 |
| ManiSkill3 | 36 | 20 | 720 |
| MuJoCo | 6 | 20 | 120 |
| MiniArcade | 19 | 10 | 190 |
| Box2D | 8 | 40 | 320 |
| RoboDesk | 6 | 20 | 120 |
| OGBench | 12 | 20 | 240 |
| Atari | 27 | 40 | 1080 |
| **Total** | **200** | – | **4020** |

# F  BASELINES

**BC.** Our multitask behavior cloning (BC) implementation closely follows that of our pretraining stage for Newt, except that it only consists of two components: the encoder $h$ and policy $p$. Both encoder and policy are conditioned on language instructions, and are trained with a BC objective that mask invalid action dimensions such that all action vectors contribute equally to the loss regardless of number of valid dimensions for that particular task. Our single-task BC policies are identical to the multitask BC policies except they are not conditioned on language instructions.

**PPO.** We base our PPO implementation off of the `cleanRL` (Huang et al., 2022b) implementation available here: `https://github.com/vwxyzjn/cleanrl/tree/master/cleanrl/ppo_continuous_action_isaacgym`, and extend it to support language-conditioning and per-task discount factors such that comparison to our method is fair. We find that the default implementation and hyperparameters performs poorly in our massively multitask RL setting, so we make a number of additional changes. Specifically, we find that *(i)* using a $\tanh$-Normal policy improves training stability, that *(ii)* increasing the number of trainable parameters from 0.5M to 3.5M leads to marginally better performance throughout training, and that *(iii)* using longer policy rollouts (128), more mini-batches and epochs (8), a smaller value loss coefficient (1), scaled down environment

rewards ($0.1\times$), and clipping of both the surrogate objective and value function. An empirical comparison of our PPO implementation before and after these changes can be found in Appendix J; we find that these changes approximately double its average score at 100M environment steps, but that both implementations perform relatively poorly compared to FastTD3 and Newt.

**FastTD3.** We base our TD3 implementation off of FastTD3 (Seo et al., 2025) which is available here: `https://github.com/younggyoseo/FastTD3`, an improved implementation of TD3 designed for the era of RL with large amounts of parallel environments. FastTD3 demonstrates robust learning across a number of different benchmarks described in Seo et al. (2025), but has not previously been evaluated in a multi-domain setting. We extend FastTD3 to support language-conditioning for compatibility with MMBench, as well as per-task discount factors to make value estimation easier across tasks. We find that $n$-step returns of $8$ are critical in our challenging multitask setting and we thus use that value by default. Appendix J shows the performance of FastTD3 for different values of $n$.

**Model-based pretraining.** This baseline is equivalent to the pretraining stage of Newt. Specifically, we pretrain all components from Equation 1 as described in Section 3.2 for a total of $200,000$ iterations, and report performance results without planning (simply using the policy prior as behavior policy) as we find it to perform the best immediately following the pretraining stage; planning eventually outperforms the policy prior but initially suffers from an inaccurate value function.

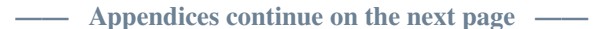

—— Appendices continue on the next page ——

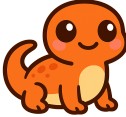

# Newt

### Appendices F−G

## G    DESCRIPTION OF PLANNING ALGORITHM

Our planning algorithm closely follows that of TD-MPC2, which uses a variant of the Cross-Entropy Method (CEM) or Model Predictive Path Integral (MPPI; (Williams et al., 2015)) for trajectory optimization with the learned world model. Specifically, TD-MPC2 plans trajectories by fitting a time-dependent multivariate Gaussian with diagonal covariance to a value landscape estimated with rollouts from the world model. Starting from an initial distribution parameterized by $(\mu^0, \sigma^0)_{t:t+H}$, $\mu^0, \sigma^0 \in \mathbb{R}^m$, $\mathcal{A} \in \mathbb{R}^m$, where $t$ is the current time step and $H$ is the planning horizon, we iteratively refit parameters $(\mu, \sigma)$ to a weighted average of the value estimates of a small set of "elite" trajectories (those whose estimates are in the top-$K$ of trajectories sampled that given iteration), and sample another set of trajectories based on the updated parameters. After a fixed number of planning iterations, the algorithm samples a trajectory from the final optimized distribution and the agent executes the first action in the returned sequence.

When evaluating our agent in a closed-loop manner, we replan trajectories at every time step and initialize $\mu^0, \sigma^0$ with the one-step shifted parameters of the previous time step (receding horizon MPC). When evaluating in an open-loop manner, the planning procedure returns the entire action sequence and we execute them one by one without replanning nor any environment feedback. To warm-start the planning procedure, TD-MPC2 generates a fixed number of action sequences by rolling out the model (in the latent space) using actions coming from the policy prior and adds them to the set of action sequences sampled at every iteration of the planning procedure. Our proposed constraint on the planning procedure early in training (when the model is less accurate than the policy prior pretrained on demonstrations, as we discuss in Section 3.2) initializes $(\mu^0, \sigma^0)$ as a linear interpolation between the action distribution coming from the policy prior $p$ and the Guassian prior used in TD-MPC2. Our annealing schedule is visualized in Figure 11. We refer the reader to Hansen et al. (2022) and Hansen et al. (2024) for more information about the planning procedure.

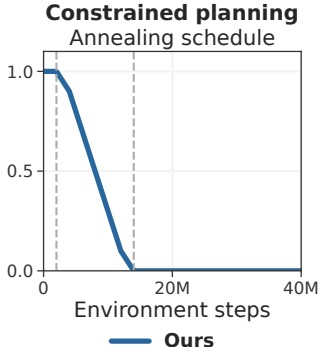

*Figure 11.* **Constrained planning.** We find that biasing planning towards the action distribution of the pretrained policy helps with exploration early in training and mitigates a drop in performance when switching from the pretrained model-free policy to planning with a world model. We use a simple linear annealing schedule (from 2M to 12M steps) as shown here.

## H  IMPLEMENTATION DETAILS

**Language embeddings.** We use text embeddings from `openai/clip-vit-base-patch32` (CLIP; (Radford et al., 2021)) which outputs continuous embeddings of 512 dimensions. Embeddings are visualized in a low-dimensional space in Figure 5.

**Image embeddings.** We use RGB image embeddings from `facebook/dinov2-base` (DINOv2; Oquab et al. (2023)) which outputs continuous embeddings of 768 dimensions. We do not use frame stacking in our experiments, but expect it to improve downstream performance as the added history information allows models to infer *e.g.* velocity information from visual observations.

**Network architecture.** All learnable components from Equation 1 are implemented as MLPs. The encoder $h$ contains a variable number of layers depending on the architecture size, and we also vary the number of $Q$-functions in our ensemble when training larger or smaller models than our default model size of 20M learnable parameters. MLPs consist of linear layers followed by LayerNorm (Ba et al., 2016) and Mish (Misra, 2019) activation functions, and the latent state representation is normalized using a simplicial embedding as originally proposed by Lavoie et al. (2023) and later applied in an RL context by Hansen et al. (2024). We summarize our Newt architecture for the default 20M parameter state-based agent with language-conditioning using a PyTorch-like notation:

```
Encoder (2,235,904 parameters): ModuleDict(
  (state): Sequential(
    (0): NormedLinear(in_features=640, out_features=1024, bias=True, act=Mish)
    (1): NormedLinear(in_features=1024, out_features=1024, bias=True, act=Mish)
    (2): NormedLinear(in_features=1024, out_features=512, bias=True, act=Simplicial)
  )
)
Dynamics (2,645,504 parameters): Sequential(
  (0): NormedLinear(in_features=1040, out_features=1024, bias=True, act=Mish)
  (1): NormedLinear(in_features=1024, out_features=1024, bias=True, act=Mish)
  (2): NormedLinear(in_features=1024, out_features=512, bias=True, act=Simplicial)
)
Reward (2,223,205 parameters): Sequential(
  (0): NormedLinear(in_features=1040, out_features=1024, bias=True, act=Mish)
  (1): NormedLinear(in_features=1024, out_features=1024, bias=True, act=Mish)
  (2): Linear(in_features=1024, out_features=101, bias=True)
)
Policy prior (2,136,096 parameters): Sequential(
  (0): NormedLinear(in_features=1024, out_features=1024, bias=True, act=Mish)
  (1): NormedLinear(in_features=1024, out_features=1024, bias=True, act=Mish)
  (2): Linear(in_features=1024, out_features=32, bias=True)
)
Q-functions (11,116,025 parameters): QEnsemble(
  (_Qs): ModuleList(
    (0-4): 5 x Sequential(
      (0): NormedLinear(in_features=1040, out_features=1024, bias=True, act=Mish)
      (1): NormedLinear(in_features=1024, out_features=1024, bias=True, act=Mish)
      (2): Linear(in_features=1024, out_features=101, bias=True)
    )
  )
)
```

When the agent is conditioned on image embeddings in addition to state and language, the encoder is expanded as follows:

```
Encoder (3,022,336): ModuleDict(
  (state): Sequential(
    (0): NormedLinear(in_features=1408, out_features=1024, bias=True, act=Mish)
    (1): NormedLinear(in_features=1024, out_features=1024, bias=True, act=Mish)
    (2): NormedLinear(in_features=1024, out_features=512, bias=True, act=Simplicial)
  )
```

and the total parameter count increases to 21M. We experiment with four model sizes: 2M parameters, 5M parameters, our default model with 20M parameters, and a larger model with 80M parameters; Table 6 details the specific architecture for each model size.

**Latent state normalization.** The simplicial embedding (Lavoie et al., 2023) is a simple method for normalization of the latent representation $\mathbf{z}$ by projecting it into $L$ fixed-dimensional simplices using a softmax operation, which was first applied in an RL context by Hansen et al. (2024). A key benefit of embedding $\mathbf{z}$ as simplices (as opposed to *e.g.* a discrete representation or squashing) is that it naturally biases the representation towards sparsity without enforcing hard constraints. Intuitively,

the simplicial embedding can be thought of as a *"soft"* variant of the vector-of-categoricals approach to representation learning proposed by Oord et al. (2017) (VQ-VAE). Whereas VQ-VAE represents latent codes using a set of discrete codes ($L$ vector partitions each consisting of a one-hot encoding), a simplicial embedding partitions the latent state into $L$ vector partitions of continuous values that each sum to 1 due to the softmax operator. This relaxation of the latent representation is akin to softmax being a relaxation of the $\arg\max$ operator. We follow Lavoie et al. (2023) and TD-MPC2 (Hansen et al., 2024) without modification and implement the simplicial normalization layer using PyTorch-like notation as follows:

```python
def simplicial(self, z, V=8):
    shape = z.shape
    z = z.view(*shape[:-1], -1, V)
    z = softmax(z, dim=-1)
    return z.view(*shape)
```

Here, `z` is the latent representation $\mathbf{z}$, and $V$ is the dimensionality of each simplex. The number of simplices $L$ can be inferred from $V$ and the dimensionality of $\mathbf{z}$. We apply a softmax to each of $L$ partitions of $\mathbf{z}$ to form simplices, and then reshape to the original shape of $\mathbf{z}$. Refer to Lavoie et al. (2023) for more details.

*Table 6.* **Model sizes.** Architecture changes that we make for our scaling experiments. *Encoder dim* is the dimensionality of each hidden layer in the state encoder, *MLP dim* is the dimensionality of hidden layers in all other parts of the agent, *Latent dim* is the dimensionality of the latent state representation, *# enc layers* is the number of encoder layers, and *Q-ensemble* is the number of $Q$-functions in the ensemble. We highlight our default model parameters in **bold**.

| Parameters | 2M | 5M | 20M | 80M |
|---|---|---|---|---|
| Encoder dim | 128 | 256 | **1024** | 2048 |
| MLP dim | 256 | 512 | **1024** | 2048 |
| Latent dim | 384 | 512 | **512** | 704 |
| # enc layers | 2 | 2 | **3** | 4 |
| Q-ensemble | 3 | 5 | **5** | 7 |

**Heuristic for discount factors.** Following TD-MPC2, we use the following heuristic to determine a suitable discount factor for tasks for which no default discount factor is available:

$$\gamma = \text{clip}(\frac{\frac{T}{5} - 1}{\frac{T}{5}}, \, [0.95, 0.995]) \tag{4}$$

where $T$ is the episode length *after* applying action repeat, and $\text{clip}$ bounds the discount factor to the interval $[0.95, 0.995]$. This heuristic ensures that tasks with short episodes are assigned a low discount factor, and vice-versa for tasks with long episodes.

**Hyperparameters.** Our hyperparameters are listed in Table 7 for the default 20M parameter agent.

*Table 7.* 🐒 **Hyperparameters.** We detail relevant hyperparameters for our method below, corresponding to our default 20M parameter agent. Our choice of hyperparameters closely follow that of TD-MPC2 (Hansen et al., 2024) with a few exceptions which we highlight in **bold**.

| Hyperparameter | Value |
| --- | --- |
| **Planning** | |
| Horizon ($H$) | 3 |
| Iterations | 6 |
| Population size | 512 |
| Policy prior samples | 24 |
| Number of elites | 64 |
| Minimum std. | 0.05 |
| Maximum std. | 2 |
| Temperature | 0.5 |
| Momentum | No |
| **Constrained planning** | **Anneal from 2M $\rightarrow$ 12M steps** |
| | |
| **Policy prior** | |
| Log std. min. | $-10$ |
| Log std. max. | 2 |
| | |
| **Replay buffer** | |
| **Capacity** | **$10,000,000$** |
| Sampling | Uniform |
| | |
| **Architecture (5M)** | |
| **Encoder layers** | **3** |
| **Encoder dim** | **1024** |
| **MLP dim** | **1024** |
| Latent state dim | 512 |
| Activation | LayerNorm + Mish |
| Number of $Q$-functions | 5 |
| Number of reward/value bins | 101 |
| Simplicial dim ($V$) | 8 |
| Simplicial temperature ($\tau$) | 1 |
| | |
| **Optimization** | |
| **Update-to-data ratio** | **0.075** |
| **Batch size** | **1024** |
| Joint-embedding coef. | 20 |
| Reward prediction coef. | 0.1 |
| Value prediction coef. | 0.1 |
| **BC coef.** | **10** |
| Temporal coef. ($\lambda$) | 0.5 |
| $Q$-fn. momentum coef. | 0.99 |
| Policy prior entropy coef. | $1 \times 10^{-4}$ |
| Policy prior loss norm. | Moving $(5\%, 95\%)$ percentiles |
| Optimizer | Adam |
| Learning rate | $3 \times 10^{-4}$ |
| Encoder learning rate | $1 \times 10^{-4}$ |
| Gradient clip norm | 20 |
| Discount factor | Heuristic |

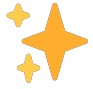

# Additional results

Appendices H−K

## I OPEN-LOOP CONTROL

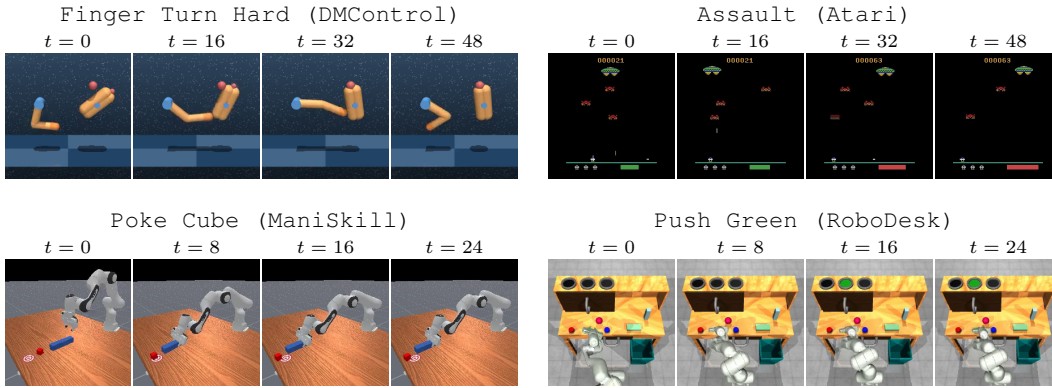

*Figure 12.* **Additional qualitative results for open-loop control.** Executing open-loop plans without any environment feedback. Our results demonstrate that Newt learns meaningful representations.

*Table 8.* **Open-loop control.** Quantitative results when planning for 48 or 24 consecutive time steps, depending on the task. We report the fraction of original closed-loop control performance achieved in an open-loop manner after 33%, 66%, and 100% of the planned time steps. Higher is better ↑.

| Task | $t = 16$ | $t = 32$ | $t = 48$ |
|------|------|------|------|
| Walker Walk | 0.68 | 0.40 | 0.25 |
| Finger Turn Hard | 0.41 | 0.39 | 0.29 |
| Assault | 0.00 | 0.10 | 0.08 |
| Lunarlander Takeoff | 0.67 | 0.40 | 0.34 |
| **Average** | **0.44** | **0.32** | **0.24** |

| Task | $t = 8$ | $t = 16$ | $t = 24$ |
|------|------|------|------|
| Point Maze | 0.88 | 0.67 | 0.37 |
| Pick Screwdriver | 0.95 | 0.89 | 0.83 |
| Poke Cube | 0.50 | 0.38 | 0.35 |
| Push Green | 0.95 | 0.89 | 0.83 |
| **Average** | **0.82** | **0.71** | **0.60** |

## J   ADDITIONAL EXPERIMENTS

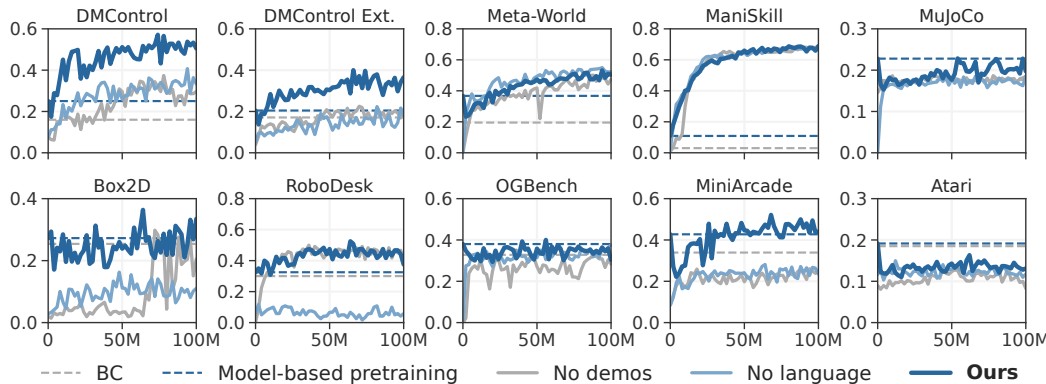

*Figure 13.* **Per-domain performance (additional baselines).** Average score of a *single* agent in each of the 10 task domains in MMBench (200 tasks). These baselines complement the results shown in Figure 14.

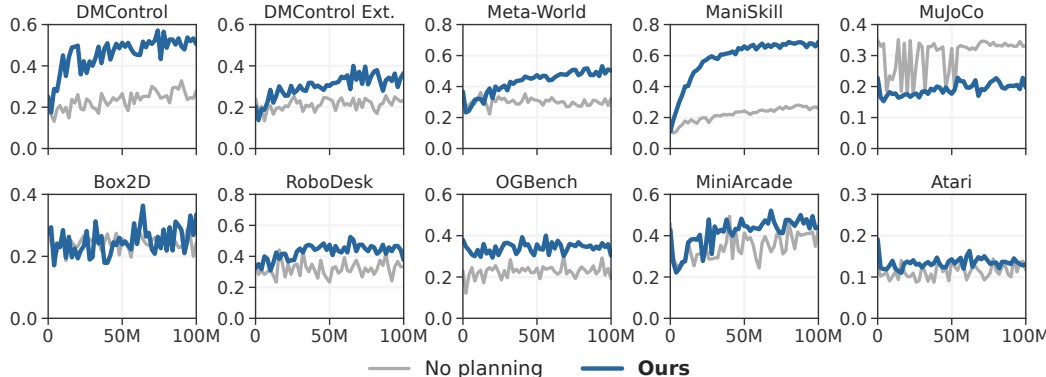

*Figure 14.* **Per-domain performance (ablation on planning).** Average score of a *single* agent in each of the 10 task domains in MMBench (200 tasks). We compare the performance of our method with and without planning with MPC. The default formulation of Newt uses planning; the baseline without planning instead uses the learned policy prior for decision-making. Our results indicate that planning plays a key role in massively multitask online RL, but that the policy prior in rare cases (MuJoCo) may outperform planning.

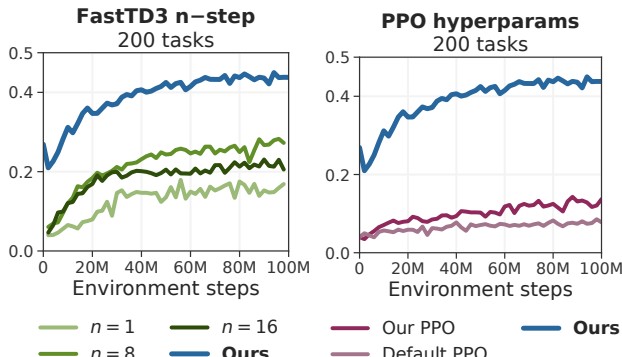

*Figure 15.* **Ablations on baselines.** We compare baseline performance across different hyperparameter configurations, and find that it has moderate impact on data-efficiency and overall performance.

*Table 9.* **Comparison to single-task TD-MPC2 agents.** Performance (normalized score; higher is better ↑) comparison between our multitask Newt agent trained for a total of 100M environment steps across all tasks (500k per task), and single-task TD-MPC2 agents trained until convergence (5M environment steps per task; 1B total steps across all tasks). We also include more fair comparisons in which the population of single-task agents have been exposed to the *same* total number of environment steps as our Newt agent. We observe that the multitask agent generally lags behind the population of single-task agents in terms of performance but that the extent to which that is the case varies drastically between task domains.

| | Task domain | Num. tasks | Multitask Newt | | Single-task TD-MPC2 | | |
|---|---|---|---|---|---|---|---|
| | | | @20M | @100M | @20M | @100M | @1B |
| | DMControl | 21 | 0.49 | 0.50 | 0.32 | 0.67 | 0.81 |
| | DMControl Ext. | 16 | 0.26 | 0.37 | 0.24 | 0.60 | 0.78 |
| | Meta-World | 49 | 0.32 | 0.50 | 0.53 | 0.70 | 0.79 |
| | ManiSkill3 | 36 | 0.52 | 0.69 | 0.36 | 0.77 | 0.92 |
| | MuJoCo | 6 | 0.18 | 0.20 | 0.35 | 0.64 | 0.82 |
| | MiniArcade | 19 | 0.40 | 0.44 | 0.50 | 0.75 | 0.94 |
| | Box2D | 8 | 0.19 | 0.33 | 0.60 | 0.87 | 0.86 |
| | RoboDesk | 6 | 0.42 | 0.38 | 0.48 | 0.67 | 0.82 |
| | OGBench | 12 | 0.29 | 0.30 | 0.43 | 0.65 | 0.93 |
| | Atari | 27 | 0.14 | 0.13 | 0.13 | 0.24 | 0.51 |
| | **Total** | **200** | **0.31** | **0.44** | **0.38** | **0.65** | **0.80** |

# K  PER-TASK RESULTS

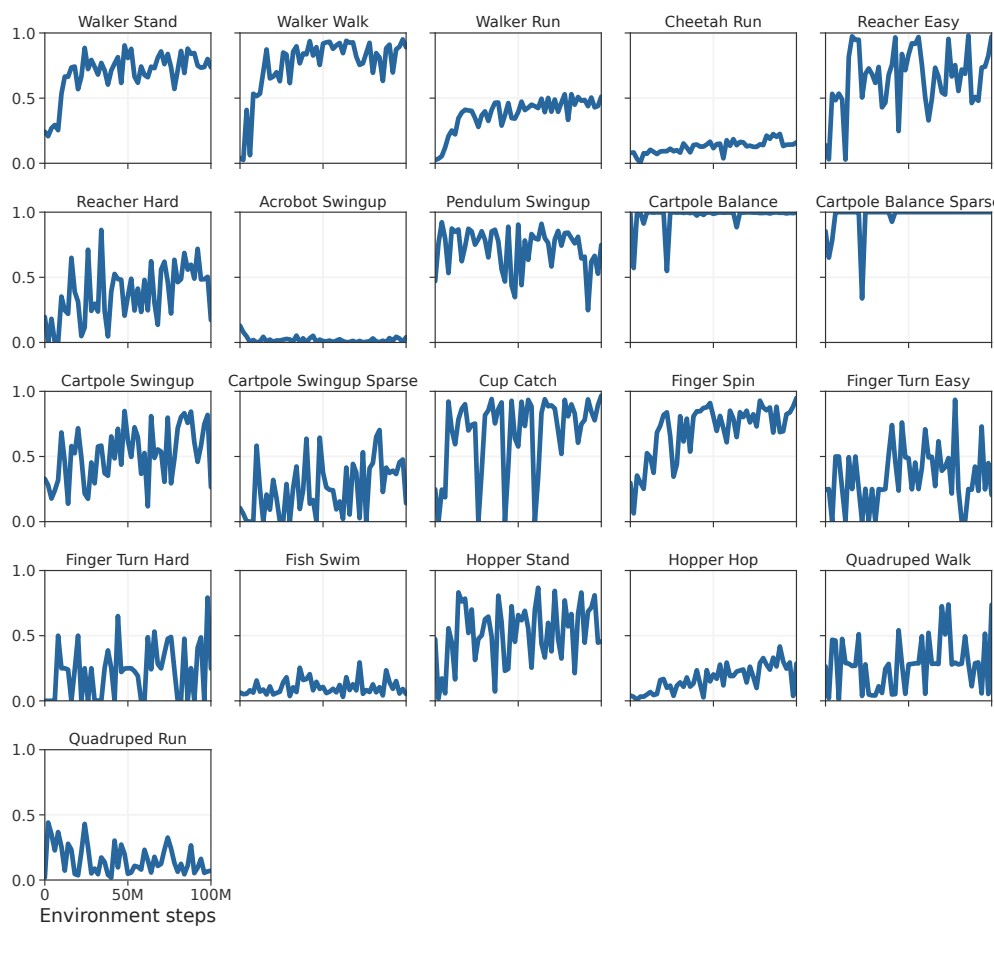

*Figure 16.* **DMControl.** Per-task normalized scores for our 20M parameter Newt agent jointly trained on all 200 tasks from MMBench. *Environment steps* refers to **total** steps across all tasks.

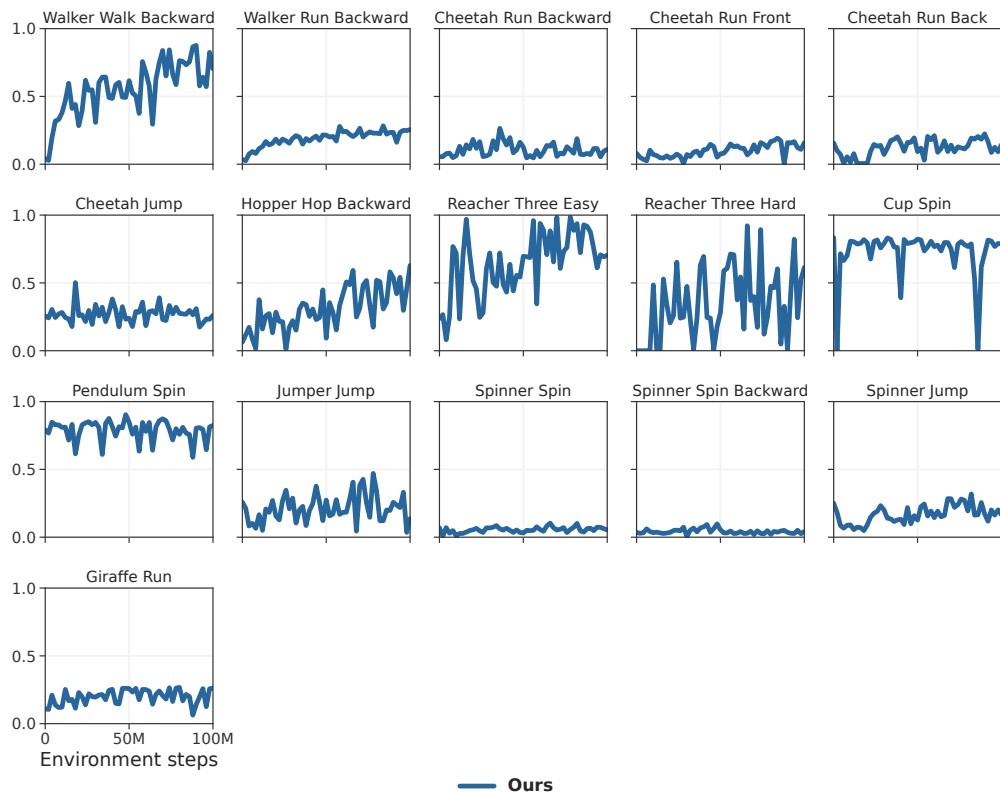

*Figure 17.* **DMControl Extended.** Per-task normalized scores for our 20M parameter Newt agent jointly trained on all 200 tasks from MMBench. *Environment steps* refers to **total** steps across all tasks.

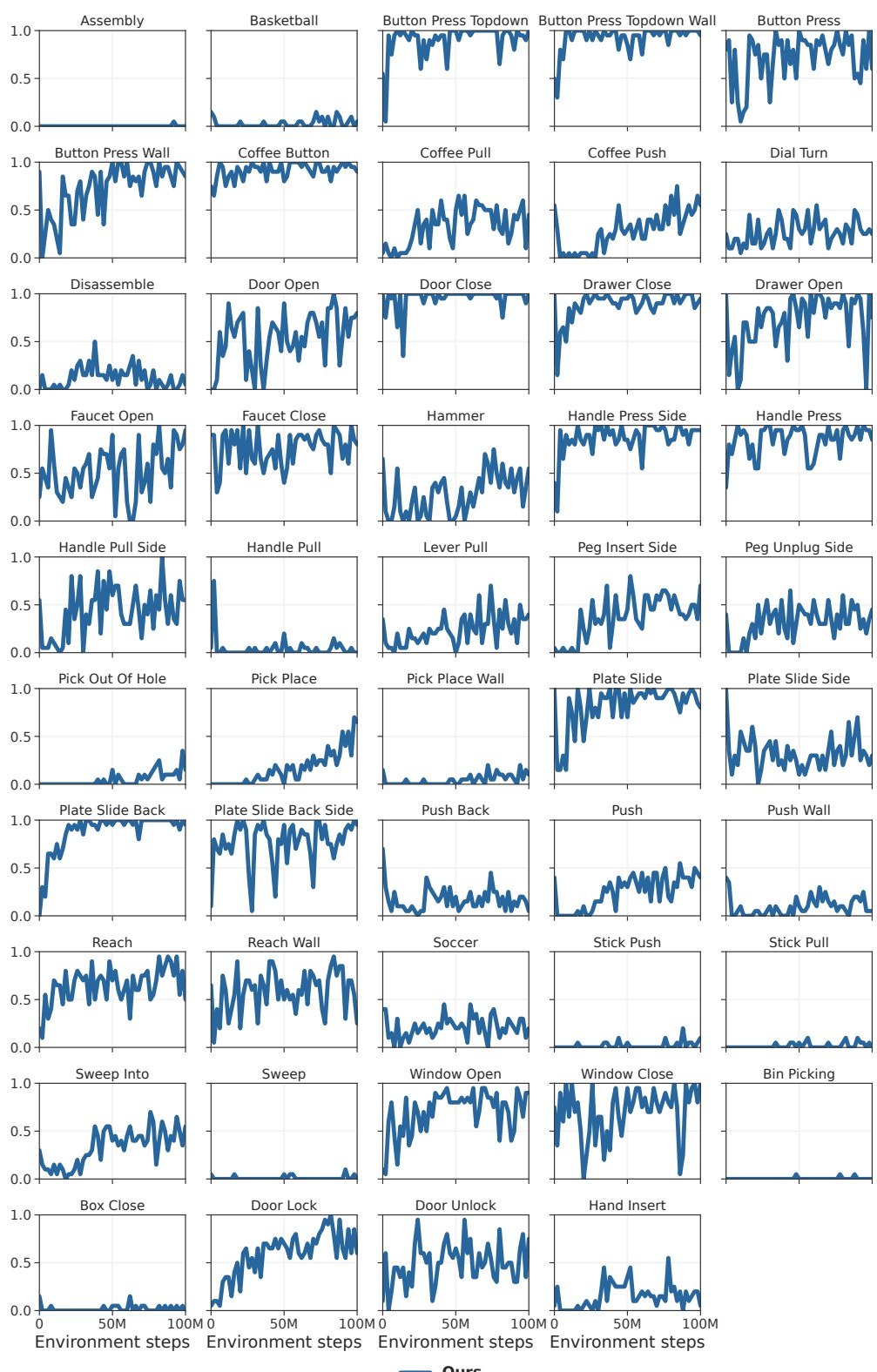

*Figure 18.* **Meta-World.** Per-task normalized scores for our 20M parameter Newt agent jointly trained on all 200 tasks from MMBench. *Environment steps* refers to **total** steps across all tasks.

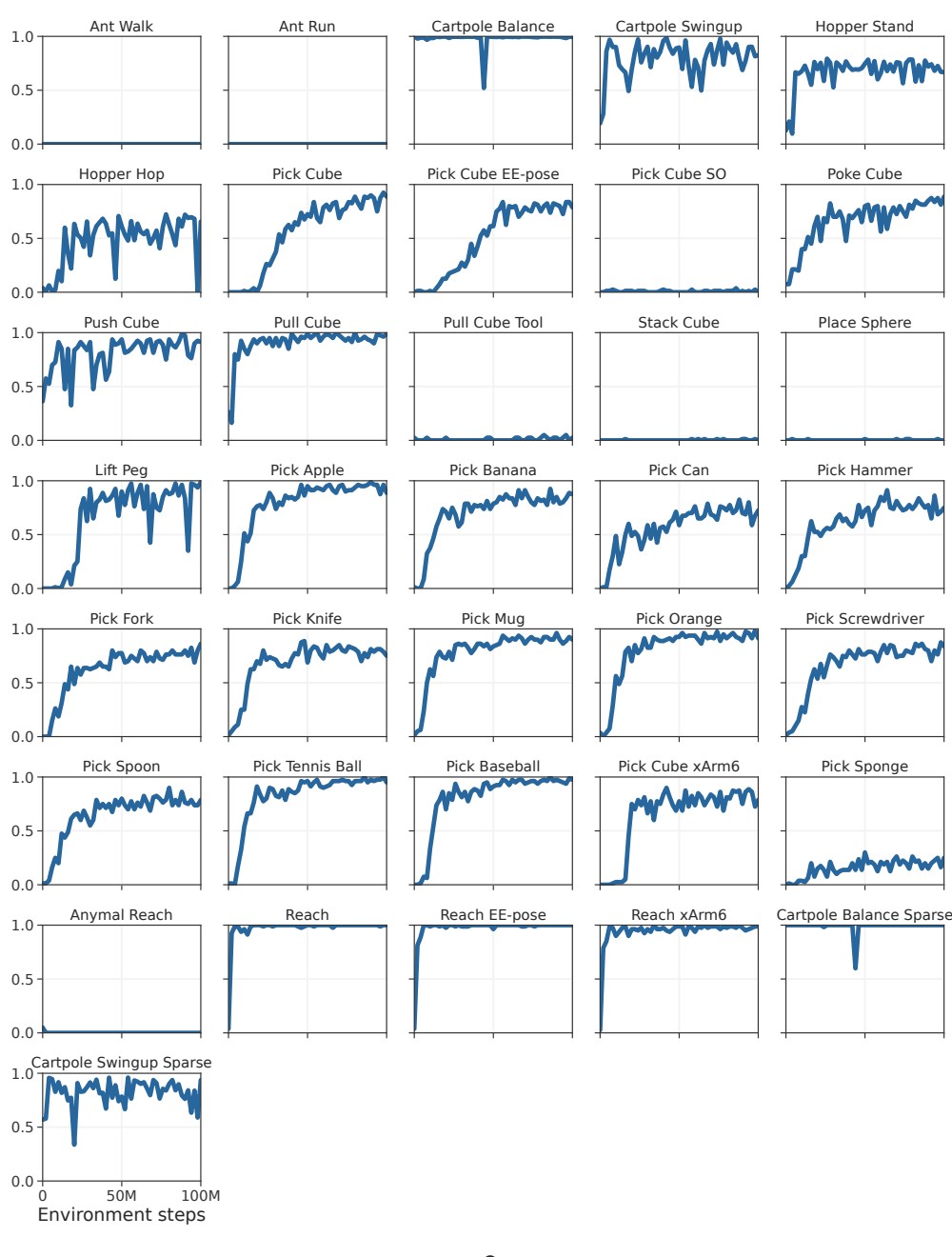

*Figure 19.* **ManiSkill.** Per-task normalized scores for our 20M parameter Newt agent jointly trained on all 200 tasks from MMBench. *Environment steps* refers to **total** steps across all tasks.

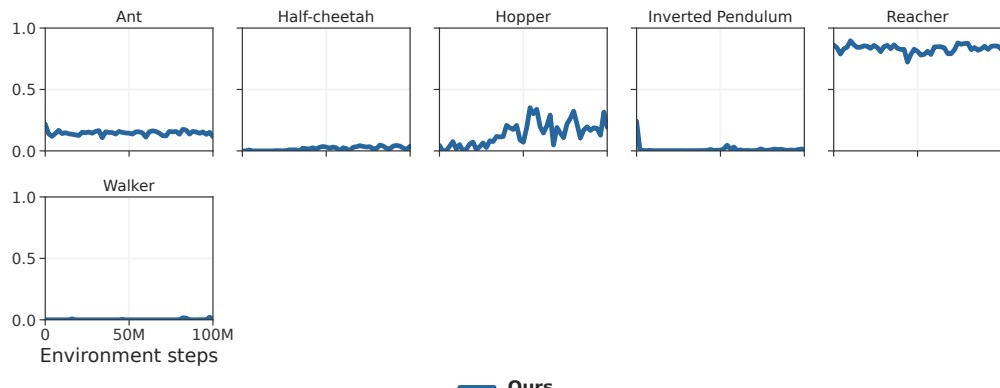

*Figure 20.* **MuJoCo.** Per-task normalized scores for our 20M parameter Newt agent jointly trained on all 200 tasks from MMBench. *Environment steps* refers to **total** steps across all tasks.

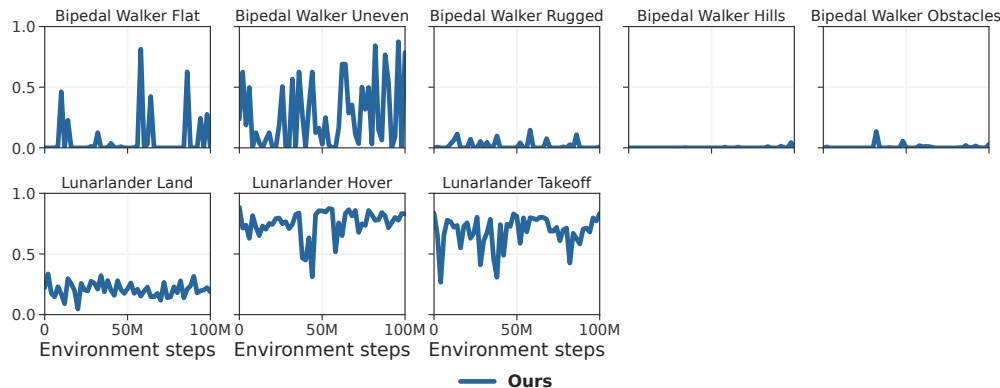

*Figure 21.* **Box2D.** Per-task normalized scores for our 20M parameter Newt agent jointly trained on all 200 tasks from MMBench. *Environment steps* refers to **total** steps across all tasks.

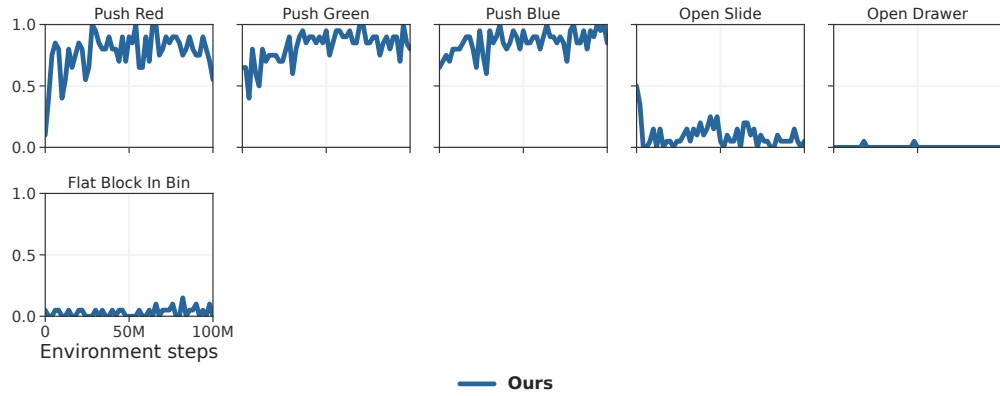

*Figure 22.* **RoboDesk.** Per-task normalized scores for our 20M parameter Newt agent jointly trained on all 200 tasks from MMBench. *Environment steps* refers to **total** steps across all tasks.

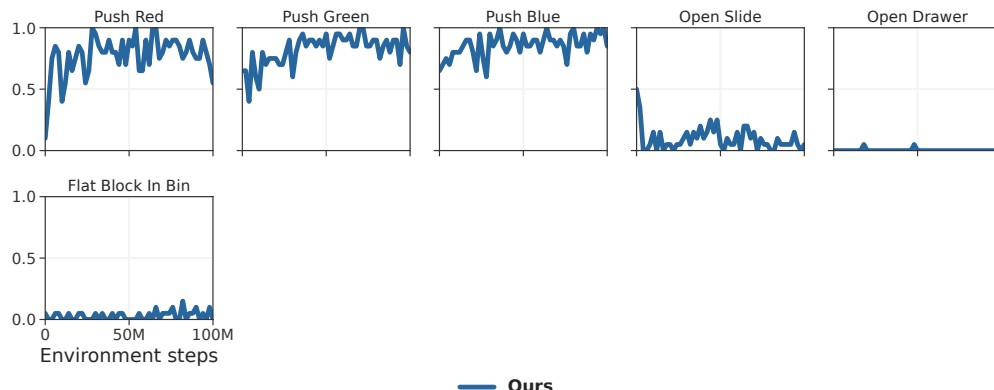

*Figure 23.* **OGbench.** Per-task normalized scores for our 20M parameter Newt agent jointly trained on all 200 tasks from MMBench. *Environment steps* refers to **total** steps across all tasks.

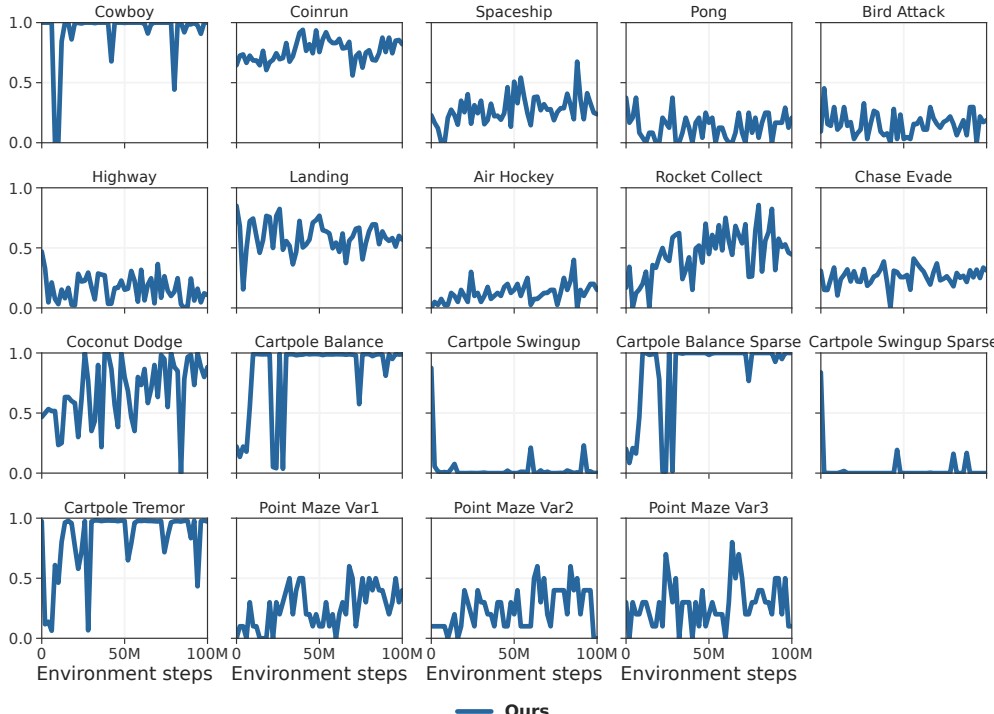

*Figure 24.* **MiniArcade.** Per-task normalized scores for our 20M parameter Newt agent jointly trained on all 200 tasks from MMBench. *Environment steps* refers to **total** steps across all tasks.

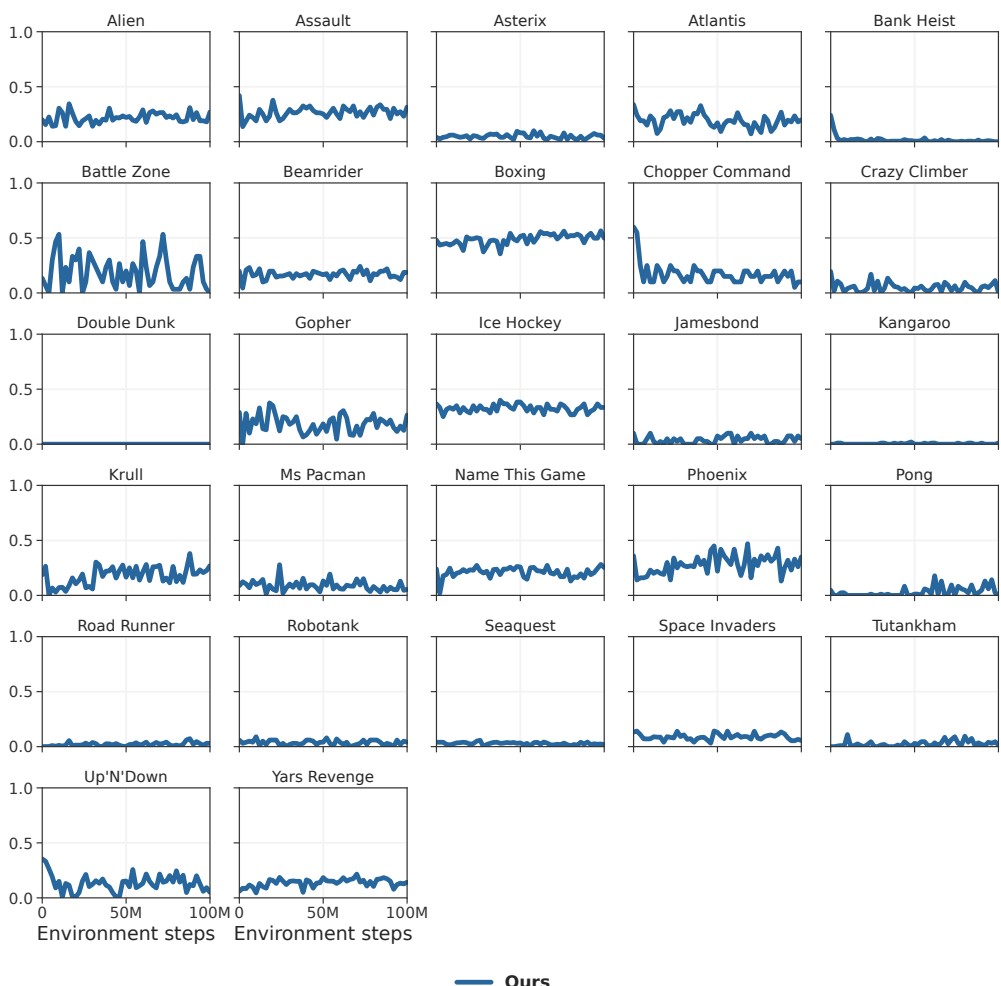

*Figure 25.* **Atari.** Per-task normalized scores for our 20M parameter Newt agent jointly trained on all 200 tasks from MMBench. *Environment steps* refers to **total** steps across all tasks.

# L    LOSS CURVES

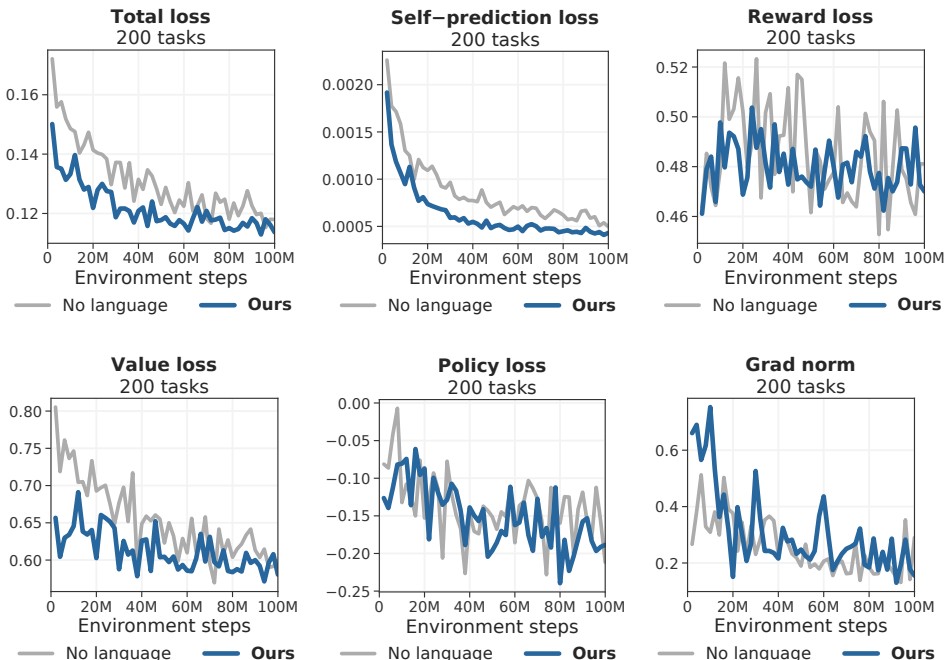

*Figure 26.* **Loss curves** for our method with and without language instructions. These curves indicate that learning is stable across both agents, but that the agent with access to language instructions generally has a lower loss during training.

