# OpenReview forum: "Learning Massively Multitask World Models for Continuous Control"
_ICLR.cc/2026/Conference — ICLR 2026 Poster_

### Official Review · Reviewer_uUxq · 2025-10-23

**Soundness:** 3
**Presentation:** 3
**Contribution:** 2
**Rating:** 8
**Confidence:** 3

**Summary:**

The paper presents an architecture to perform online RL on a bigger scale, training one policy for many different tasks. The learning algorithm requires as a first step a behavioural cloning approach to warm-start the network and make the exploration problem simpler. It then performs online RL, outputting actions after an MPC planning step trough a world model. It hence combines model-based and model-free RL approaches.

**Strengths:**

The paper presents good results on continous  control benchmarks which is still an interesting problem. Especially training one policy over this big variety of tasks is interesting to see. The algorithm is presented clearly and is easy to follow, and the model-based MPC aspect of it is interesting. I really appreciate the effort of the authors of making the code and the checkpoints accessible, this makes it possible to reproduce the results and build on-top of them.

**Weaknesses:**

My major concern is the applicability of this to real-world continuous control problems. While in simulation the results look good, it requires over 100M steps to train this policy which would be unfeasable on a real-world application. I also think the paper would benefit from ablating the usefulness of the different components - specifically interesting would be to understand how useful is the mpc planning is, how much learning of the world model helps performance as well as  how much does the initial bc training helps.

**Questions:**

how useful is the mpc planning?
how much does learning of the world model helps performance?
how much does the initial bc training help?

---

> ### Author Response · Authors · 2025-11-21
> **Thank you!**
>
> We thank the reviewer for their valuable feedback. We address your comments in the following.
>
> ----
> **Q1:** My major concern is the applicability of this to real-world continuous control problems. While in simulation the results look good, it requires over 100M steps to train this policy which would be unfeasable on a real-world application.
>
> **A1:** We agree that running 100M online interaction steps on physical hardware would be impractical with today’s real-robot setups. However, our goal in this work is not to propose a training protocol that is executed end-to-end on physical robots, but rather to answer a more fundamental question: can online RL, in principle, be scaled to hundreds of tasks at once? In that context, 100M steps spread over 200 tasks corresponds to roughly 0.5M steps per task, which is comparable to or *smaller* than what is commonly used for single-task continuous-control benchmarks. Moreover, the intended usage pattern is analogous to foundation models in other domains: a large multitask policy is trained once (in simulation and/or with access to a large demonstration dataset), and then reused and fine-tuned in data-scarce real-world settings rather than retrained from scratch for every deployment. Our results show that such a multitask policy can indeed be obtained with online RL, and that it supports rapid adaptation to new tasks and embodiments (Table 1 and Figure 9), which is precisely the regime relevant for e.g. real robots. We have made this intended framing clearer in the paper: MMBench and Newt are designed for research and prototyping of massively multitask agents, not as a prescription for how many real-world interaction steps a physical robot must collect for multitask capabilities to emerge. We have clarified this in Section 2.0 of our revised paper.
>
> ----
> **Q2:** I also think the paper would benefit from ablating the usefulness of the different components - specifically interesting would be to understand how useful is the mpc planning is, how much learning of the world model helps performance as well as how much does the initial bc training helps.
>
> **A2:** This is a valid point! We already conduct a total of 16 ablation experiments for Newt in the submitted version of the paper (primarily pertaining to batch size, model size, language and image inputs, pretraining, demonstrations; can be found in Figure 8, Table 2, Appendix I), as well as additional ablations on baseline design choices and hyperparameters in Appendix I (Additional Experiments). However, we agree that your suggested ablations would be informative to readers:
> - *“how much does the initial bc training helps”*: This ablation is already provided in Figure 8, and Appendix I further shows the benefit of pretraining on a per-domain basis. In short, we find that the addition of pretraining achieves higher asymptotic performance overall (we believe this may be due to more effective exploration as a result of pretraining), as well as higher data efficiency in the early stages of training (this is to be expected).
> - *“how useful is the mpc planning is”*: We did not do this for the original submitted version but we agree that it would be useful to quantify! We have added this ablation to Appendix I of our revised paper ([screenshot](https://i.imgur.com/NSAYf8f.png)). Our results show that disabling MPC planning and instead relying solely on the learned policy prior for decision-making leads to almost stagnant learning curves: it achieves a normalized score of 0.27 at 100M environment steps vs. 0.25 immediately after pretraining (across all 200 tasks). For comparison, we achieve a score of 0.44 (@100M) when using planning.
> - *“how much learning of the world model helps performance”*: We agree that this is an interesting research question but find it non-trivial to ablate: if we remove the world model altogether, this also removes our ability to do MPC which evidenced by the previous ablation mostly fails to learn.
>
> ----
> We believe that our response and additional experiment results address your concerns. However, please do not hesitate to let us know if you have any additional comments.
>
> **(1/1)**

---

> > ### Author Response · Authors · 2025-11-26
> > **Reminder**
> >
> > Dear reviewer uUxq,
> >
> > This is a gentle reminder that the rebuttal period is slowly coming to an end. We would really appreciate it if you could take a moment to read through our response and let us know if you have any additional comments. Thank you for your time!

---

> > ### Comment · Reviewer_uUxq · 2025-11-27
> >
> > thank you to the authors for addressing my comments and for providing the additional ablation. I will keep my score as it is. thank you.

---

### Official Review · Reviewer_73Ch · 2025-10-29

**Soundness:** 3
**Presentation:** 4
**Contribution:** 2
**Rating:** 4
**Confidence:** 5

**Summary:**

This paper tackles the scalability challenge of online RL for continuous control by introducing MMBench, a large-scale benchmark spanning 200 diverse tasks across 10 domains (e.g., DMControl, Meta-World, Atari, and a newly proposed MiniArcade). Each task includes language instructions, demonstrations, and multi-modal observations. The authors also propose Newt, a language-conditioned multitask world model built upon TD-MPC2. Newt first pretrains on demonstrations to obtain task-aware representations and action priors, then performs online optimization across tasks with architectural refinements and action supervision. Experiments show that Newt outperforms several baselines (BC, PPO, FastTD3) in multitask performance and exhibits generalization to unseen tasks.

**Strengths:**

The paper has several notable merits:
- MMBench provides a unified framework across 10 heterogeneous domains with consistent data handling and language-conditioned tasks.
- The paper introduces reasonable design choices such as discrete regression for reward/value prediction and per-task discount factors, with comprehensive ablations supporting their impact.
- The paper is well-organized, figures effectively illustrate key results, and open-sourced resources (200+ checkpoints, 4000+ demos) significantly enhance reproducibility.

Overall, the paper represents a solid step toward scalable, general-purpose control systems.

**Weaknesses:**

### 1. Novelty concerns in core contributions

While MMBench contains 200 tasks, most of them are directly inherited from existing benchmarks (e.g., DMControl, Meta-World). This limits its novelty compared to benchmarks such as ManiSkill3, which introduces fundamentally new task paradigms. Similarly, Newt—although incorporating CLIP/DINOv2 encoders and demonstration conditioning—builds incrementally upon TD-MPC2, without a clear paradigm shift.

### 2. Missing analysis of task scalability

The paper emphasizes scaling to hundreds of tasks but lacks quantitative analysis on scaling behavior. For example, performance is not evaluated as the task count increases (e.g., 50 → 100 → 200 tasks), leaving unclear whether multitask training indeed benefits from more tasks. Moreover, since MMBench spans disjoint domains (e.g., Atari vs. Meta-World), the paper should analyze cross-domain transfer—whether training on visually distinct domains interferes with or enhances learning in others—and include ablations comparing full multitask vs. domain-specific training.

### 3. Incomplete baselines in MBRL

Experimental comparisons primarily focus on model-free RL and behavioral cloning baselines, while omitting competitive model-based RL counterparts. In particular, DreamerV3, which is explicitly designed for multitask continuous control, is absent, as is a multitask-adapted TD-MPC2 baseline (used for demo collection). Without these, it is difficult to attribute performance gains to the proposed architectural innovations rather than inherited advantages from TD-MPC2.

### 4. Insufficient evaluation under state-limited conditions

Most evaluations assume access to full low-dimensional states, which is unrealistic in real-world control settings. The paper lacks experiments under state-limited or purely visual conditions (e.g., partial observations or agent-only states), which would better demonstrate robustness and practical applicability.

**Questions:**

1. The paper employs masking to handle inconsistent action/state dimensions. Could masking lead to sparse gradient updates or optimization inefficiencies for high-dimensional action spaces (e.g., humanoid control with 50+ joints)? Have you explored task-specific action embeddings or shared latent action representations as scalable alternatives?

2. Table 3 reports Newt’s training time, but it is unclear whether this includes the cost of training 200 single-task TD-MPC2 agents used for demonstration generation. Please clarify whether the total computation time includes both stages (demo collection + multitask training) to ensure fair comparison and full cost transparency.

---

> ### Author Response · Authors · 2025-11-21
> **Thank you!**
>
> We thank the reviewer for their valuable feedback. We address your comments in the following.
>
> ----
> **Q1:** While MMBench contains 200 tasks, most of them are directly inherited from existing benchmarks (e.g., DMControl, Meta-World). This limits its novelty compared to benchmarks such as ManiSkill3, which introduces fundamentally new task paradigms. Similarly, Newt—although incorporating CLIP/DINOv2 encoders and demonstration conditioning—builds incrementally upon TD-MPC2, without a clear paradigm shift.
>
> **A1:** We respectfully disagree with the statement that MMBench is *“just”* a collection of existing tasks. While it is true that many tasks are adapted from DMControl, Meta-World, Atari, etc., our contribution with MMBench is to *standardize and compose* 200 diverse tasks into a single benchmark explicitly designed for massively multitask online RL: all tasks share a unified API, conventions, language instructions, and demonstrations, and can be trained jointly by a single agent. In addition, we introduce **41 new tasks**, half of which constitute an entirely new task domain (MiniArcade). Benchmarks such as ManiSkill3 are complementary: they introduce a collection of robotics tasks within a single simulator, whereas MMBench focuses on breadth across simulators, embodiments, reward structures, and task categories, which is precisely the regime required to stress-test generalist control policies and multitask world models. There are, to our knowledge, no such benchmarks available today besides MMBench. Secondly, we also take issue with the statement that Newt *“builds incrementally upon TD-MPC2, without a clear paradigm shift”*. Our algorithmic changes include: *(1)* adding language and vision inputs, *(2)* supervised pretraining on demonstrations, *(3)* BC-regularized policy prior, *(4)* constrained planning, and *(5)* support for environment parallelization across hundreds of online RL tasks each with distinct observation and action spaces. While each of these changes may seem incremental in isolation, we see them as pragmatic and important steps towards a broader paradigm shift in online RL: **MMBench defines the first large-scale multitask setting for online RL, and Newt provides concrete evidence that an agent can successfully be trained with online RL in this setting**.
>
> ----
> **Q2:** The paper emphasizes scaling to hundreds of tasks but lacks quantitative analysis on scaling behavior. For example, performance is not evaluated as the task count increases (e.g., 50 → 100 → 200 tasks), leaving unclear whether multitask training indeed benefits from more tasks.
>
> **A2:** We agree that this is a very interesting question, but refrained from including such experiments as the selection / subsampling of tasks at different task counts (as well as the choice of performance metric) would naturally have a large influence over the results and conclusions when our task domains are relatively disjoint. If you subsample by task domain (i.e., training on 2 -> 6 -> 10 domains), you will likely see an improvement in training performance (normalized score) with fewer domains, and transfer performance would be increasingly dependent on the specific choice of transfer tasks (i.e., how similar they are to the smaller set of training tasks); this would leave significant room for bias in the experimental results. If you instead subsample uniformly over tasks, you run into a different but related issue: the resulting subsets mix tasks with very different difficulty, observability, reward structure, and domain characteristics. Apparent “scaling trends” as we go from 50 -> 100 -> 200 tasks would then be dominated by which specific hard/easy tasks happen to be included, rather than any intrinsic effect of the task count. For example, excluding a small number of particularly challenging tasks could make a 50-task subset look strictly better than a 200-task set, even though this is purely an artifact of task choice. This is further amplified by the choice of performance metric (per-task average, domain-weighted average, training vs. transfer performance etc.), which could qualitatively change the conclusion.
> Given these confounding factors, we designed MMBench to have a single, fully specified 200-task training set and evaluate all methods under this standardized condition. We do however compare a multitask BC policy to 200 single-task BC policies which could be considered a special case of the scaling question above that avoids sampling bias in its entirety, and we have now also added comparisons to multitask TD-MPC2 (partial results; still running; [screenshot](https://i.imgur.com/DLg0BK9.png)) as well as 200 single-task TD-MPC2 agents ([screenshot](https://i.imgur.com/ytwY0kT.png)) in Figure 7 and Appendix I, respectively, of our revised paper. That said, we agree that carefully controlled scaling studies are a valuable research direction enabled by our benchmark and we hope to see more of that in the future.
>
> **(1/3)**

---

> > ### Author Response · Authors · 2025-11-21
> >
> > **Q3:** the paper should analyze cross-domain transfer—whether training on visually distinct domains interferes with or enhances learning in others—and include ablations comparing full multitask vs. domain-specific training.
> >
> > **A3:** This is a great suggestion and definitely something we will be happy to add. As mentioned above, we already compared multitask BC vs. 200 single-task BC policies, and we have now added the equivalent comparison for RL as well: multitask Newt vs. 200 single-task TD-MPC2 policies. We would also like to emphasize that we already include zero-shot and few-shot transfer experiments in the paper (Table 1 and Figure 9, respectively) which addresses the question of transfer to new tasks/embodiments within multiple task domains seen in training. Your question pertains specifically to transfer *across* domains which too is a very interesting research question, but not something that we can feasibly run during the rebuttal period. We expect to see *positive* transfer across domains that feature similar embodiments and tasks (e.g. Meta-World and ManiSkill) and *interference* across domains that have little in common (e.g. Meta-World and Atari). We are committed to adding a comparison between our 10-domain Newt and 10 domain-specific Newt agents (similar to our single-task baselines) for the camera-ready version, which we believe addresses your question about cross-domain transfer.
> >
> > ----
> > **Q4:** Experimental comparisons primarily focus on model-free RL and behavioral cloning baselines, while omitting competitive model-based RL counterparts. In particular, DreamerV3, which is explicitly designed for multitask continuous control, is absent, as is a multitask-adapted TD-MPC2 baseline (used for demo collection).
> >
> > **A4:** We agree that a multitask TD-MPC2 baseline is a very reasonable baseline to include, as is a comparison to the 200 single-task TD-MPC2 agents we previously used solely for demonstration data collection for MMBench (the RL equivalent of our 200 single-task BC agents baseline). We have added both of these baselines to the revised version of our paper, in Figure 7 ([screenshot](https://i.imgur.com/DLg0BK9.png)) and Appendix I (Additional Experiments; [screenshot](https://i.imgur.com/ytwY0kT.png)), respectively; thank you for this suggestion. It is true that DreamerV3 could also be an interesting method to compare to, but *(i)* numerous papers have found DreamerV3 to be inferior to TD-MPC2 and model-free methods like FastTD3 (which we do compare to) and SAC in single-task continuous control tasks, and *(ii)* we would like to push back on the reviewer’s claim that *“In particular, DreamerV3, which is explicitly designed for multitask continuous control, is absent”* since **DreamerV3 is in fact not designed for multitask continuous control**: only a small portion of their experiments are continuous control tasks (DMControl), and none of their experiments can be considered multitask. For these reasons, we believe that focusing on more competitive baselines that previously have been applied in a multitask continuous control setting (such as TD-MPC2, FastTD3, PPO, and BC) is more informative to readers.
> >
> > ----
> > **Q5:** Most evaluations assume access to full low-dimensional states, which is unrealistic in real-world control settings. The paper lacks experiments under state-limited or purely visual conditions (e.g., partial observations or agent-only states), which would better demonstrate robustness and practical applicability.
> >
> > **A5:** This is a valid point, but the premise of massively multitask online RL directly on real robots is itself currently impractical if not infeasible: before tackling things like perception, hardware constraints, and safety, we first need to understand whether online RL even *scales* to hundreds of tasks in a controlled setting. For this reason, we deliberately focus the main experiments on low-dimensional states (fairly common in RL for continuous control and far more accessible for researchers with limited resources), and which isolates the multitask, multi-embodiment RL and world modeling aspects of our work from confounding perception issues. That said, MMBench fully supports visual observations and we already include some limited evaluations that compare state only vs. state + vision performance across domains, and we would love to see more work of this nature in the future; that is precisely the type of research questions that our benchmark enables the community to address!
> >
> > **(2/3)**

---

> > > ### Author Response · Authors · 2025-11-21
> > >
> > > **Q6:** The paper employs masking to handle inconsistent action/state dimensions. Could masking lead to sparse gradient updates or optimization inefficiencies for high-dimensional action spaces (e.g., humanoid control with 50+ joints)? Have you explored task-specific action embeddings or shared latent action representations as scalable alternatives?
> > >
> > > **A6:** We have not explored more advanced methods such as learned (latent) action representations, but that is a very interesting open research direction that we believe MMBench is uniquely suited for. One could imagine that a learned action representation would map e.g. all end-effector position control action spaces to similar points in the latent space irrespective of the specific robot embodiments, and similarly group Atari actions based on game mechanics. We believe that our language instructions already allow for this type of action space modeling to emerge since they explicitly describe the embodiment and action space in text, but a more systematic exploration of learned action spaces certainly would be interesting as a future research project. We have added a few sentences to Section 5 (Conclusion) of the revised version of our paper that explicitly mentions this as a potential limitation and open research question.
> > >
> > > ----
> > > **Q7:** Table 3 reports Newt’s training time, but it is unclear whether this includes the cost of training 200 single-task TD-MPC2 agents used for demonstration generation. Please clarify whether the total computation time includes both stages (demo collection + multitask training) to ensure fair comparison and full cost transparency.
> > >
> > > **A7:** We agree that fair comparison and cost transparency is important, which is why we *(i)* detail hardware requirements + training wall-time for various hardware configurations in Section 4 / Table 3, *(ii)* release our full code, demonstration data, and 200+ model checkpoints to reviewers and the public, and *(iii)* provide 22 pages of appendices with detailed information about our MMBench benchmark tasks and demonstrations, Newt agent and baseline implementations, our efforts in tuning hyperparameters for the FastTD3 and PPO baselines, additional experimental results and ablations, loss curves, learning curves for every single task, and much more. We agree that it would be informative to also provide compute estimates for the single-task TD-MPC2 agents that we trained during development of MMBench; we have expanded Appendix D (Demonstrations) to include additional information (highlighted in green) about demonstration collection, which is now available in the revised version of our paper ([screenshot](https://i.imgur.com/htBE2R3.png)). That said, we would like to reiterate that our contributions include both MMbench and Newt, and emphasize that we consider training of single-task expert policies as well as demonstration data collection strictly a part of the development of MMBench, since Newt does not leverage these expert policies (we simply release them since future work might find them useful) and has no specific requirements wrt. the source of demonstrations used for pretraining; our results in Figure 1 even show that removing demonstrations altogether still yields a stronger multi-task agent than existing methods FastTD3 and PPO on our proposed MMBench task set.
> > >
> > > ----
> > > We believe that our response and additional experiment results address your concerns. However, please do not hesitate to let us know if you have any additional comments.
> > >
> > > **(3/3)**

---

> > > > ### Comment · Reviewer_73Ch · 2025-11-22
> > > >
> > > > The authors’ responses and revisions have resolved most of my concerns. While I remain somewhat uncertain about the novelty of the proposed method, I acknowledge that the dataset introduced by the authors is highly valuable for the RL community. The additional experiments on model-based RL approaches and the clearer explanations of the work’s technical details make the revised paper more thorough and solid. Consequently, I am willing to raise my score.

---

> > > > > ### Author Response · Authors · 2025-11-22
> > > > > **Thank you!**
> > > > >
> > > > > Thank you for the quick response! We are glad to hear that our changes and additional experiments address most of your concerns and that you are willing to raise your score accordingly. If you have any additional questions please do not hesitate to let us know! Either way, we appreciate you taking the time to read our rebuttal :-)

---

### Official Review · Reviewer_nJ1K · 2025-10-30

**Soundness:** 3
**Presentation:** 2
**Contribution:** 3
**Rating:** 6
**Confidence:** 3

**Summary:**

This paper proposes MMBench, a multitask RL benchmark containing 200 different control tasks across 10 domains. TDMPC2 agents on each single task are trained to collect expert demonstrations, while both the model checkpoints and demonstrations are open-source. In addition, this paper substitute language descriptions for task indices for better distinguish among tasks, which makes Newt,  a multi-task world model based on TD-MPC2. Experiments show comparable results with a strong baseline FastTD3.

**Strengths:**

1. This paper proposes a benchmark which integrates domains that are popularly studied in the RL community, releasing single-task checkpoints and dataset which is significant not only for multi-task RL but also offline, O2O and continuous RL research.
2. The empirical results show advantages over baselines on ManiSkill and DMControl.
3. Model info such as training time, model architecture is detailed presented.
4. The figures are well drawn and easy to understand.

**Weaknesses:**

1. There is no preliminary description so the problem setting confuses me at the beginning. In the multi-task RL setting a task label $n$ should be added to the original $(s, a, s', r)$ but in Line.275 there is only $(s, a, r)$.

2. Newt only shows performance boost over baselines in DMC and Maniskill out of all 10 domains. In Meta-World, MuJoCo, Box2D, Robodesk and Atari it's just on par with FastTD3, while in OGBench and MiniArcade it's on par with behavior cloning.

3. Selected baselines are not strong enough. FastTD3 is a strong baseline and in its original paper is compared with strong model-based baselines such as TDMPC2 and Dreamerv3, but it mainly reports results on humanoidbench, mujoco playground and Issaclab, neither of them are included in MMBench. Moreover, as long as Newt is built upon TDMPC2, it surprises me that TDMPC2 is not listed as a baseline in this paper.

**Questions:**

1. To make it clear, is there only one agent being trained to interact with all 200 environments and collect online trajectories?

2. What does the "Language Instructions: None" refer to in Line 399-405? Is there a task index provided for each trajectory?

3. Is there any results that support the claim in Line 147 that one-hot encoding limits the potential for transferring to unseen tasks?

---

> ### Author Response · Authors · 2025-11-21
> **Thank you!**
>
> We thank the reviewer for their valuable feedback. We address your comments in the following.
>
> ----
> **Q1:** To make it clear, is there only one agent being trained to interact with all 200 environments and collect online trajectories?
>
> **A1:** Yes, correct. We train a **single** agent via **online interaction on 200 diverse tasks** in parallel. This is, to our knowledge, the **first** time that such a milestone has been achieved in the RL community.
>
> ----
> **Q2:** There is no preliminary description so the problem setting confuses me at the beginning
>
> **A2:** This is a valid point. It was our intention for the presentation of the MMBench benchmark in Section 2 to serve that purpose, but we recognize that the problem setting can be stated more explicitly. We have revised Section 2 to explicitly formalize the problem setting ([screenshot](https://i.imgur.com/4hBNF9l.png)), but are very open to reviewer suggestions on alternative ways that this could be achieved.
>
> ----
> **Q3:** Newt only shows performance boost over baselines in DMC and Maniskill out of all 10 domains. In Meta-World, MuJoCo, Box2D, Robodesk and Atari it's just on par with FastTD3, while in OGBench and MiniArcade it's on par with behavior cloning.
>
> **A3:** We would like to clarify two things that are mostly orthogonal:
> 1. Our contributions are twofold: *(i)* we propose MMBench, the first benchmark for massively multitask RL. We observe that the RL community is increasingly interested in multitask learning as opposed to the traditional single-task online RL setting but researchers so far have been limited to small-scale problem settings (e.g. Meta-World) in part because there is a lack of benchmarks and infrastructure in place for training, and in part because of a common belief that online RL does not scale which we believe has made people hesitant to even try. MMbench unifies lots of existing tasks as well as 41 new tasks, complete with training infrastructure, demonstrations, and expert policies for future research in this area. This paves the way for our second contribution: *(ii)* we propose Newt, a method that extends the TD-MPC2 model-based RL algorithm to the massively multitask online RL setting. While TD-MPC2 has been shown to work well in both single-task online RL settings and multi-task (up to 80 tasks) offline RL, we demonstrate that it can be extended to **multi-task online RL with 200 tasks** spanning a much more diverse range of environments and embodiments than previously, e.g. **10 task domains vs. just 2 in the TD-MPC2 paper**. We firmly believe that this paradigm shift in how RL is used as well as a concrete benchmark for exploring such ideas is in itself a very valuable contribution.
> 2. Newt outperforms **all** multi-task RL baselines in 4 task domains (DMControl, DMControl Extended, ManiSkill, MiniArcade), as well as the multi-task BC baseline in 8 task domains (DMControl, DMControl Extended, Meta-World, ManiSkill, MuJoCo, Box2D, RoboDesk, MiniArcade), and performs significantly better on average as evidenced by Figure 1. While there certainly still is room for improvement in performance, that is exactly our intention with proposing such a challenging benchmark; if Newt was able to fully learn all 200 tasks as is, the benchmark would have relatively little value to the research community.
>
> **(1/3)**

---

> > ### Author Response · Authors · 2025-11-21
> >
> > **Q4**: Selected baselines are not strong enough. FastTD3 is a strong baseline and in its original paper is compared with strong model-based baselines such as TDMPC2 and Dreamerv3, but it mainly reports results on humanoidbench, mujoco playground and Issaclab, neither of them are included in MMBench. Moreover, as long as Newt is built upon TDMPC2, it surprises me that TDMPC2 is not listed as a baseline in this paper.
> >
> > **A4:** We agree that selection of strong baselines is important. However, given that we are the first to explore online RL in a massively multitask setting spanning so many domains, there are regrettably very few readily available baselines to compare against. We choose to adapt FastTD3 to our problem setting since it is a very recent method for continuous control and, although they do not conduct multitask experiments in their paper, their official codebase does support multitask RL and it details suggested hyperparameters for multitask RL on Meta-World which is one of our 10 task domains. To make the FastTD3 more competitive in our multi-domain setting we further conduct our own hyperparameter tuning for which we report the results in Appendix I (Additional Experiments) to ensure full transparency and reproducibility. We similarly do this for the PPO baseline as well. It is true that DreamerV3 could also be an interesting method to compare to, but (i) numerous papers have already found DreamerV3 to be inferior to TD-MPC2 and model-free methods like FastTD3 and SAC in single-task continuous control tasks (including the FastTD3 paper referenced in your comment), and (ii) we would like to emphasize that applying DreamerV3 to **multitask continuous control** is an open research problem: only a small portion of the experiments in the DreamerV3 paper are continuous control tasks (DMControl), and none of their experiments can be considered multitask. For these reasons, we believe that focusing on more competitive baselines that previously have been applied in a multitask continuous control setting (such as TD-MPC2, FastTD3, PPO, and BC) is more informative to readers. That said, we agree with the sentiment of your comment that “it surprises me that TDMPC2 is not listed as a baseline” and have now added both a multitask TD-MPC2 baseline (partial results; still running; [screenshot](https://i.imgur.com/DLg0BK9.png)) as well as a comparison to 200 single-task TD-MPC2 agents (the RL equivalent of our multitask and single-task BC policy baselines; [screenshot](https://i.imgur.com/ytwY0kT.png)); results for these methods have been added to Figure 7 and Appendix I, respectively, in the revised version of our paper, as well as an additional ablation on Newt that disables planning and instead uses only the learned policy prior (also Appendix I; [screenshot](https://i.imgur.com/NSAYf8f.png)). Thank you for your suggestion!
> >
> > ----
> > **Q5:** Is there any results that support the claim in Line 147 that one-hot encoding limits the potential for transferring to unseen tasks?
> >
> > **A5:** Intuitively, the performance ceiling when representing tasks via language vs. one-hot encoding should be the same when considering only in-domain (training) tasks since both allow the agent to distinguish tasks assuming the language instructions are unique to each task. However, it is not clear how one-hot encoding could be used to represent a new task since it was never observed in training: do you expand the one-hot encoding dictionary by 1 (200 -> 201) in which case the embedding conveys no information about the new task? Or do you assign the new task the same one-hot encoding as one of the training tasks in which case there still is a mismatch between the task at hand and the task indicator? We have updated the relevant paragraph in the paper to make this limitation more clear, and also include an additional ablation on this in Figure 8 of our revised paper to verify this intuition experimentally; we provide more details on this in the following answer.
> >
> > **(2/3)**

---

> > > ### Author Response · Authors · 2025-11-21
> > >
> > > **Q6:** What does the "Language Instructions: None" refer to in Line 399-405? Is there a task index provided for each trajectory?
> > >
> > > **A6:** This particular ablation removes access to language instructions without replacing it with something else, so the ablation directly quantifies how much information is conveyed with language vs. the observations alone. However, we agree that an additional comparison to Newt with task indices (i.e. one-hot encoding / learnable task embeddings as used in the original TD-MPC2 paper) is useful. We have added this new ablation to Figure 8 of our revised paper ([screenshot](https://i.imgur.com/qfQKaDk.png)); the experiment is still running so results are currently incomplete, but we observe performance on the training tasks to be mostly the same as when conditioning on language. This makes intuitive sense based on our answer to the previous question (one-hot encoding can perfectly distinguish tasks from each other during training but it is unclear how one would represent an unseen task), but we agree that validating it in this manner is useful to readers.
> > >
> > > ----
> > > We believe that our response and additional experiment results address your concerns. However, please do not hesitate to let us know if you have any additional comments.
> > >
> > > **(3/3)**

---

> > > > ### Author Response · Authors · 2025-11-26
> > > > **Reminder**
> > > >
> > > > Dear reviewer nJ1K,
> > > >
> > > > This is a gentle reminder that the rebuttal period is slowly coming to an end. We would really appreciate it if you could take a moment to read through our response and let us know if you have any additional comments. Thank you for your time!

---

### Author Response · Authors · 2025-11-21
**General comment**

**We thank all reviewers for their thoughtful comments, and have revised our manuscript based on your feedback; the list of changes are available below. We have also responded to your individual comments.**

**Summary of revisions:** We summarize all major changes to our manuscript below; these changes have also been highlighted (green) in the new version.
- ([nJ1K](https://openreview.net/forum?id=MPabX9LEds&noteId=W8pvGCVoVm)) Added a subsection (2.1) which formally defines our problem setting of massively multitask online RL ([screenshot](https://i.imgur.com/4hBNF9l.png)).
- ([nJ1K](https://openreview.net/forum?id=MPabX9LEds&noteId=W8pvGCVoVm), [73Ch](https://openreview.net/forum?id=MPabX9LEds&noteId=55C1gPr24b)) Added a multitask TD-MPC2 baseline to Figure 7 ([screenshot](https://i.imgur.com/DLg0BK9.png)) as well as a comparison to 200 single-task TD-MPC2 agents in Appendix I ([screenshot](https://i.imgur.com/ytwY0kT.png)). We show that Newt learns more reliably than the naive TD-MPC2 baseline, which is also supported by the numerous ablations on our algorithmic improvements shown in Figure 8 ([screenshot](https://i.imgur.com/qfQKaDk.png)).
- ([nJ1K](https://openreview.net/forum?id=MPabX9LEds&noteId=W8pvGCVoVm)) Added an additional ablation to Figure 8 on task conditioning that replaces language embeddings with task indices as in the original TD-MPC2 paper ([screenshot](https://i.imgur.com/qfQKaDk.png)). We show that performance is mostly the same on training tasks with the caveat that task indices provide no mechanism for generalization to new tasks.
- ([73Ch](https://openreview.net/forum?id=MPabX9LEds&noteId=55C1gPr24b)) Added further details to Appendix D (Demonstrations) on the computational cost of demonstration data collection ([screenshot](https://i.imgur.com/htBE2R3.png)).
- ([uUxq](https://openreview.net/forum?id=MPabX9LEds&noteId=ZyO9oP7m6d)) Added a Newt ablation that disables MPC planning and instead relies on the learned policy prior for decision-making ([screenshot](https://i.imgur.com/NSAYf8f.png)). We show that planning is a key factor in the success of Newt.

Some of our new baselines and ablations are still running, so you will find that a small number of curves are incomplete; we will update our figures when the training runs complete but expect the conclusions to remain the same. Note that we have also made a number of smaller improvements to our revised manuscript based on your feedback; we address these changes directly in our responses to each reviewer.

Again, we thank the reviewers for their constructive feedback. We believe that all comments have been addressed in this revision, but are happy to address any further comments from reviewers.

Best,

*Authors of Newt*

---

### Author Response · Authors · 2025-12-04
**Post-rebuttal summary**

Dear ACs, SACs, PCs, and reviewers,

Thank you all for your efforts in reviewing during this somewhat unusual review period. To save you some time, we'd like to summarize key aspects of our paper and rebuttal:
- **Contributions.** Our contributions are twofold: *(i)* we propose **MMBench**, the first benchmark for massively multitask RL (200 tasks spanning 10 task domains and many more embodiments). MMbench unifies lots of existing tasks as well as 41 new tasks, complete with training infrastructure, demonstrations, and expert policies for future research in this area. This paves the way for our second contribution: *(ii)* we propose **Newt**, a method that extends the TD-MPC2 model-based RL algorithm to the massively multitask online RL setting. We train a **single** agent via **online interaction on 200 diverse tasks** in parallel.
- **Changes.** We list the main changes that we made to address reviewer feedback in a [general comment](https://openreview.net/forum?id=MPabX9LEds&noteId=MZ70oqZ9b9), which includes: several new baselines (multitask TD-MPC2, 200 single-task TD-MPC2 agents) and ablations (no planning, one-hot task ID instead of language), a new subsection that formally defines the problem setting, as well as additional info about demo collection in appendix D. We also make a series of smaller changes which are highlighted in the revised PDF.
- **Reviewer responses.** In response to our rebuttal, reviewer `73Ch` decided to raise their score from `4 -> 6` and reviewer `uUxq` chose to maintain their score of `8`. Unfortunately, we were not able to get a response from reviewer `nJ1K` (score `6`) before the discussion period concluded unexpectedly, but we believe that their concerns have been addressed.

Thanks again for your time and effort in reviewing. We really appreciate it!

---

### Meta-Review · Area_Chair_DWWB · 2025-12-24

**Summary:**

This paper introduces MMBench, a large-scale benchmark for continuous control comprising 200 tasks across 10 diverse domains, and proposes Newt, a scalable multi-task world model based on TD-MPC2. The reviewers recognized the significant contribution of the benchmark to the community, particularly the unification of diverse tasks with language instructions and the release of extensive datasets (checkpoints and demonstrations). The primary concerns informing the decision initially focused on the sufficiency of baselines (specifically the absence of a direct comparison with TD-MPC2 given the method builds upon it), the variability of Newt's performance gains across different domains, and questions regarding the formal problem definition. Following the rebuttal, the inclusion of the requested baselines and additional ablations significantly strengthened the paper. I recommend Accepting the paper as it provides valuable infrastructure for future research in massive multitask RL.

**Reviewer Concerns:**

Addressed Concerns:

•	A critical concern was the lack of comparison against the base algorithm TD-MPC2 in a multitask setting. The authors successfully addressed this by adding a Multitask TD-MPC2 baseline and a comparison against 200 Single-task TD-MPC2 agents, demonstrating Newt's improvements are not merely inherited.

•	The reviewer noted a lack of preliminary description for the specific multitask setting. The authors added a formal Problem Formulation subsection in the revision to clarify the setting.


•	Reviewers requested clearer ablations on the utility of planning and task conditioning. The authors added a "No planning" ablation and a "Task ID vs. Language" ablation, quantitatively verifying the design choices.

•	Confirmed that a single agent is indeed trained on all 200 environments simultaneously.

Outstanding Concerns:

•	Reviewer expressed concern that the 100M training steps are infeasible for real robots. While the authors clarified that the goal is a simulation-based foundation model recipe (pretrain in sim -> finetune real), the high sample complexity remains a barrier for direct real-world application without simulation.

**Reviewer Scores:**

•  Reviewer uUxq: This reviewer was highly positive about the benchmark's scale and the clear presentation. They explicitly stated in the post-rebuttal discussion that they would keep their score as 8, noting that the authors successfully addressed their comments regarding ablations.

•  Reviewer nJ1K: This reviewer initially gave a score of 6. Their main critique was the absence of a TD-MPC2 baseline and confusion regarding the problem setup. Since the authors directly addressed these points by adding the requested Multitask/Single-task TD-MPC2 baselines and a formal problem definition section, it is highly probable that nJ1K would have raised their score.

•  Reviewer 73Ch: This reviewer explicitly stated an intention to raise their score following the rebuttal, acknowledging the authors' clarifications.

---

### Decision · Program_Chairs · 2026-01-26

Accept (Poster)